# A Survey of Reasoning in Autonomous Driving Systems: Open Challenges and Emerging Paradigms

**Kejin Yu**[1*]                                             *ykj21@mails.tsinghua.edu.cn*

**Yuhan Sun**[2*]                                         *51285902023@stu.ecnu.edu.cn*

**Taiqiang Wu**[3*]                                               *takiwu@connect.hku.hk*

**Ruixu Zhang**[1]                                     *rx-zhang25@mails.tsinghua.edu.cn*

**Zhiqiang Lin**[1]                                         *linzq24@mails.tsinghua.edu.cn*

**Yuxin Meng**[1]                                     *meng-yx25@mails.tsinghua.edu.cn*

**Junjie Wang**[1†]                                          *wangjunjie@sz.tsinghua.edu.cn*

**Yujiu Yang**[1]                                         *yang.yujiu@sz.tsinghua.edu.cn*

[1] *Tsinghua University*   [2] *East China Normal University*   [3] *The University of Hong Kong*

**Reviewed on OpenReview:** *https://openreview.net/forum?id=XwQ7dc4bqn*

## Abstract

The development of high-level autonomous driving (AD) is shifting from perception-centric limitations to a more fundamental bottleneck, namely, a deficit in robust and generalizable reasoning. Although current AD systems manage structured environments, they consistently falter in long-tail scenarios and complex social interactions that require human-like judgment. Meanwhile, the advent of large language and multimodal models (LLMs and MLLMs) presents a transformative opportunity to integrate a powerful cognitive engine into AD systems, moving beyond pattern matching toward genuine comprehension. However, a systematic framework to guide this integration is critically lacking. To bridge this gap, we provide a comprehensive review of this emerging field and argue that reasoning should be elevated from a modular component to the system's cognitive core. Specifically, we first propose a novel Cognitive Hierarchy to decompose the monolithic driving task according to its cognitive and interactive complexity. Building on this, we further derive and systematize seven core reasoning challenges, such as the responsiveness-reasoning trade-off and social-game reasoning. Furthermore, we conduct a dual-perspective review of the state-of-the-art, analyzing both system-centric approaches to architecting intelligent agents and evaluation-centric practices for their validation. Our analysis reveals a clear trend toward holistic and interpretable "glass-box" agents. In conclusion, we identify a fundamental and unresolved tension between the high-latency, deliberative nature of LLM-based reasoning and the millisecond-scale, safety-critical demands of vehicle control. For future work, a primary objective is to bridge the symbolic-to-physical gap by developing verifiable neuro-symbolic architectures, robust reasoning under uncertainty, and scalable models for implicit social negotiation.

---

*Equal contributions. Kejin Yu and Yuhan Sun contributed to this work as research assistants at Tsinghua University.
†Corresponding authors.

# 1   Introduction

> *"The eye sees only what the mind is prepared to comprehend."*

Robertson Davies (1952)

Autonomous driving (AD) aims to build a transportation system that is safer, more efficient, and more accessible (Muhammad et al., 2020). A major bottleneck in autonomous driving systems is shifting from the physical limitations of perception and control systems to a deficit in robust and generalizable reasoning. This challenge manifests in scenarios requiring intricate situational understanding and commonsense, such as navigating temporary traffic control (Ghosh et al., 2024) or compensating for perception system degradation (Matos et al., 2025). However, integrating reasoning capabilities into AD systems remains underexplored (Plebe et al., 2024). To bridge the gap, we provide a comprehensive view of the integration of **large language and multimodal models (LLMs and MLLMs)** as a cognitive engine to address these **reasoning deficits in AD systems**.

For AD systems, the central and remaining challenge is shifting from perception to reasoning, specifically for large-scale real-world deployments (Chen et al., 2021). As reported by Waymo and Cruise, a more fundamental challenge for AD is the lack of advanced reasoning, such as "planning discrepancy" and "prediction discrepancy" (Boggs et al., 2020). With the development of hardware, the bottleneck no longer lies in perceptual capability, but, *in absence of an integrated reasoning framework* (Chen et al., 2021; Mahmood & Szabolcsi, 2025; Xu & Sankar, 2024). Therefore, the current primary objective of AD is to develop a cohesive cognitive architecture that enables advanced reasoning across these individual components.

Fortunately, LLMs and MLLMs exhibit remarkable reasoning capabilities, providing a promising solution to the AD reasoning bottleneck. Through pre-training on large-scale data, these LLMs and MLLMs exhibit powerful emergent reasoning abilities (Webb, 2023; Yin et al., 2024) and can further leverage a rich repository of commonsense knowledge for complex situational assessment (Zhang et al., 2024a; Wei et al., 2024b). Although their internal mechanisms differ fundamentally from human cognition, they can effectively simulate deliberative thought processes and thus address the reasoning deficit in AD (Xu et al., 2025b). As shown in Fig. 1, the reasoning capacity enables AD systems to react appropriately in challenging scenarios. For instance, when a ball rolls onto a street, a system with such reasoning can infer that an unseen child may follow and thus prompt the vehicle to decelerate preemptively. Such a capacity for probabilistic, context-aware inference is crucial for navigating the inherent unpredictability of real-world environments, and its profound potential as a new cognitive architecture motivates the systematic review presented in this survey.

To systematically analyze the central role of reasoning in autonomous driving and establish a clear foundation for the subsequent discussion, this survey makes the following key contributions:

- **Articulation of the Core Reasoning Deficit in AD.** We systematically position the integration of reasoning as the central and unsolved challenge for next-generation autonomous systems. We move beyond broad AI-in-driving surveys to specifically articulate why LLM-based reasoning is essential for overcoming documented real-world failures (e.g., mishandling novel construction zones or misinterpreting human social cues), establishing a focused foundation for the review.
- **A Novel Cognitive Hierarchy for Driving.** We propose a new conceptual framework to deconstruct the monolithic driving task based on cognitive and interactive complexity. This hierarchy comprises three distinct levels: (1) the **Sensorimotor Level** (vehicle-to-environment), (2) the **Egocentric Reasoning Level** (vehicle-to-agents), and (3) the **Social-Cognitive Level** (vehicle-in-society). This framework provides a principled methodology for analyzing the required reasoning capabilities at each layer.
- **A New Taxonomy of Core Reasoning Challenges.** Building on this cognitive hierarchy, we derive and systematize seven key challenges that impede the deployment of LLM-based reasoning in AD. Our framework allows us to analyze how these challenges (e.g., Responsiveness-Reasoning Tradeoff, the Social Game) manifest with different priorities at each cognitive level, providing a structured problem space for the research community.

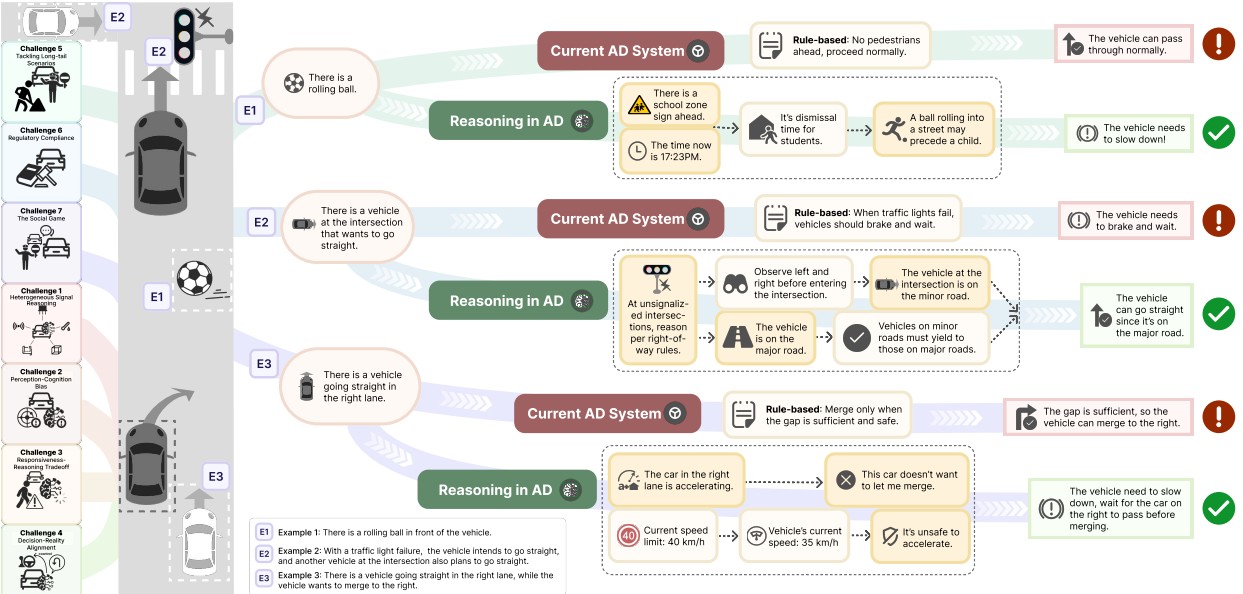

Figure 1: Motivation: why explicit reasoning matters in autonomous driving. The left panel summarizes seven recurring reasoning challenges in our taxonomy. The right panel presents three illustrative scenarios (E1–E3) that contrast a representative *current AD system* driven by rule-based heuristics with a *reasoning-in-AD* approach that integrates contextual signals, traffic rules, and multi-agent interaction cues via explicit inference (dashed boxes). The comparison highlights how brittle policies can yield unsafe or overly conservative actions (red), while structured reasoning supports context-appropriate decisions (green).

- **A Dual-Perspective Taxonomy and Analysis of the State-of-the-Art.** We provide a comprehensive review of current research through a structured analysis that distinguishes between the intelligent agent and its validation. We explore the field from two complementary viewpoints: (1) **system-centric approaches** and (2) **evaluation-centric practices**. Our analysis reveals a clear trend toward holistic, interpretable "glass-box" agents but identifies a critical gap in methods for verifying their real-time safety and social compliance.

The detailed structure of this survey is presented as follows and summarized in Fig. 2. Sec. 2 provides the preliminary concepts of reasoning paradigms within LLMs and MLLMs. Sec. 3 introduces our new cognitive hierarchy to deconstruct the autonomous driving task. Building on this framework, Sec. 4 details our taxonomy of the seven core reasoning challenges that must be addressed. We then conduct a comprehensive review of the state-of-the-art from two perspectives: Sec. 5 surveys system-centric approaches, analyzing current architectures and methodologies, while Sec. 6 reviews evaluation-centric practices, covering the critical benchmarks and datasets. Finally, Sec. 7 concludes the survey by summarizing key findings and proposing promising directions for future research.

## 2 Preliminary: Reasoning in LLMs and MLLMs

Reasoning, a cornerstone of human intelligence, refers to the systematic process of forming conclusions or decisions from evidence and prior knowledge. It is integral to cognitive functions such as problem-solving, decision-making, and critical analysis. In the domain of artificial intelligence, enhancing the reasoning capabilities of large language models (LLMs) and multimodal large language models (MLLMs) represents a critical step in the pursuit of artificial general intelligence (AGI) (Huang & Chang, 2023; Huang & Zhang, 2024; Han et al., 2025; Yan et al., 2025; Qu et al., 2025; Liu et al., 2025a; Zhang et al., 2024c). This advancement allows models to transition from pattern recognition and instruction execution to more complex problem-solving. Specifically for MLLMs, reasoning facilitates the coherent integration of information from

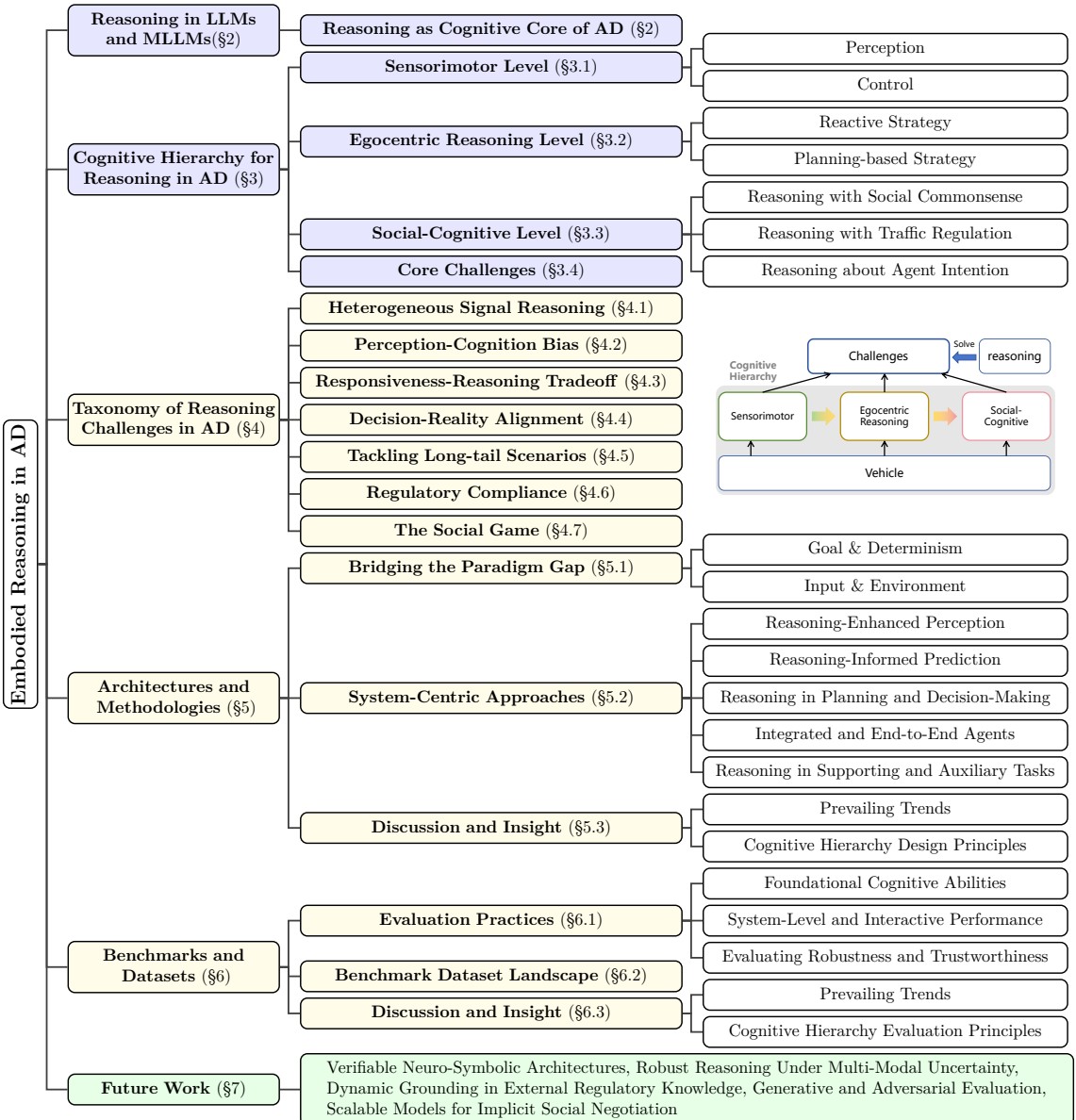

Figure 2: The outline of the survey on reasoning in autonomous driving systems.

diverse modalities, such as vision and text, to form a unified understanding (Wang et al., 2024d; 2025c; Li et al., 2025b; Shou et al., 2025). For specific reasoning paradigms of LLMs and MLLMs, please refer to the Appendix A.

## A New Paradigm: Reasoning as the Cognitive Core of Autonomous Driving

Traditional AD systems often rely on a modular pipeline that consists of perception, prediction, planning, and control (Huang & Chen, 2020). While this architecture demonstrates success in structured environments, it suffers from fundamental limitations. These limitations include significant information loss between discrete modules (Zhou et al., 2024), overreliance on predefined rules, and inherent brittleness when encountering uncertain or ambiguous scenarios.

We do not claim that perception and control have been fully solved. Instead, we emphasize that in complex and long-tail scenarios, insufficient reasoning ability can amplify perception errors, rule conflicts, and inter-

action uncertainty, thereby becoming one of the key constraints for reliable real-world deployment of AD systems.

This survey explores a new paradigm that moves beyond the traditional pipeline. We propose that reasoning should not be treated as another sequential module but instead should be elevated to the role of a cognitive core for the entire system (Xu et al., 2024b). The function of this core is not to replace existing modules but rather to understand, coordinate, and empower them. This approach transforms the system from a rigid, linear process into an integrated, intelligent agent capable of holistic scene comprehension and adaptive decision-making. To fully appreciate the motivation for this paradigm shift, we contrast it with preceding frameworks.

Classical rule-/logic-based planning and decision systems (Bhuyan et al., 2024), while strong in formal specification, can be brittle and difficult to scale to the dynamic, unstructured nature of real-world driving, partly due to the growing complexity of rules and exceptions in open environments (Bouchard et al., 2022). Causal inference models, though theoretically robust in distinguishing correlation from causation, often face substantial practical challenges due to the immense complexity and the presence of unobserved confounding variables inherent in traffic scenarios, making comprehensive causal modeling difficult to operationalize at scale (Zhang et al., 2020). Similarly, pre-LLM neuro-symbolic systems, which attempted to merge neural perception with symbolic logic, were hampered by significant integration challenges, often requiring laborious manual design of interfaces and domain-specific languages (Sun et al., 2020), while still facing the symbol grounding problem (Harnad, 1990).

These limitations suggest the need for a more scalable reasoning interface. In this context, we view LLM-based reasoning not as "symbolic" reasoning in the classical sense but as *language-mediated, learned, explicit deliberation* that helps coordinate modules and handle rule conflicts and interactions in complex driving settings.

## 3 A Cognitive Hierarchy for Reasoning in Autonomous Driving

A monolithic view of "driving" is insufficient for enabling targeted intervention. To map the advanced reasoning capabilities of a central Cognitive Core to concrete operational challenges, a granular deconstruction of the driving task itself is required. Drawing inspiration from the evolving complexity of the interaction between the vehicle, its environment, and human agents (Wilde, 1976; Wang et al., 2022), we propose a new conceptual framework to hierarchically structure driving tasks. As shown in Fig. 3, this framework comprises three distinct levels: (1) the Sensorimotor Level, (2) the Egocentric Reasoning Level, and (3) the Social-Cognitive Level. This hierarchy provides a principled methodology to deconstruct the monolithic concept of "driving" into layers of increasing cognitive and interactive complexity.

### 3.1 Sensorimotor Level

This level corresponds to the most fundamental operations of driving and representing atomic actions. These tasks typically involve a direct mapping from a perceptual input to an executive output and require minimal complex decision-making. The core capabilities at this level are illustrated by the following representative examples:

- **Perception (Cognition & Sensing).** Perception refers to the identification of elements in the environment and the determination of their states. This category includes foundational modules for object detection and vehicle localization, analogous to a human driver visually recognizing another vehicle or a traffic signal.
- **Control (Cognition & Actuation).** Control pertains to the execution of direct physical commands. In an autonomous system, this involves carrying out basic vehicle commands for steering, acceleration, and braking, similar to a human driver responding to a simple directive to press a pedal or turn the steering wheel.

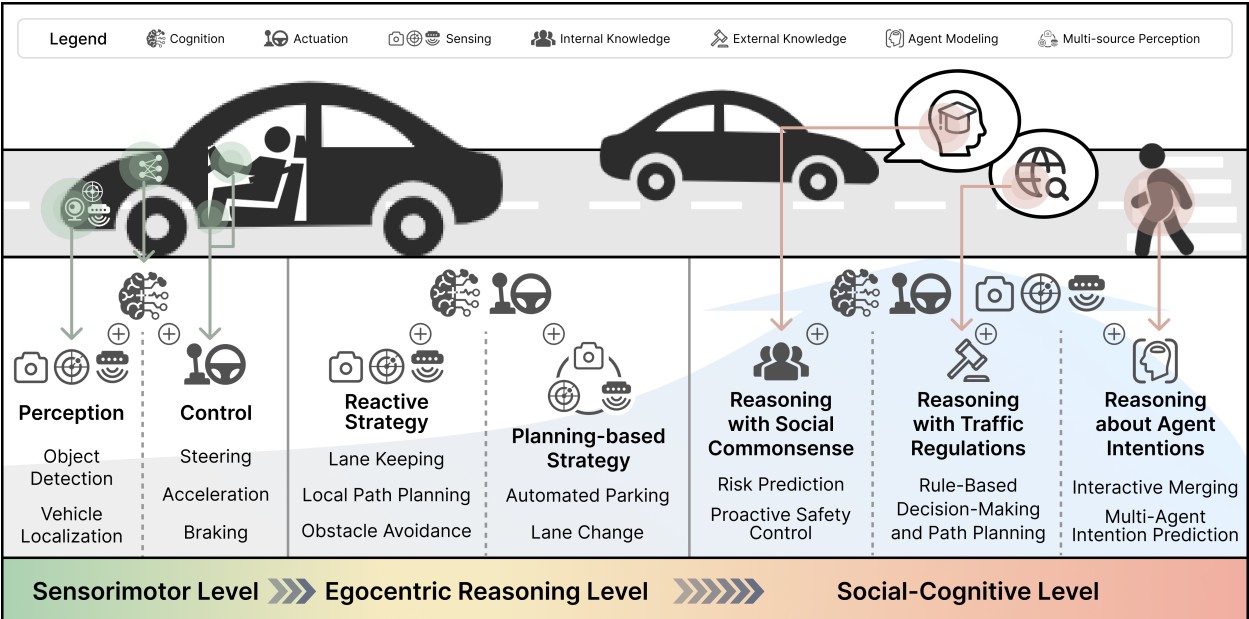

Figure 3: The proposed Cognitive Hierarchy for reasoning in autonomous driving. This framework deconstructs the monolithic "driving" task into three distinct levels of increasing cognitive and interactive complexity: (1) the Sensorimotor Level, (2) the Egocentric Reasoning Level, and (3) the Social-Cognitive Level.

## 3.2 Egocentric Reasoning Level

At this level, the system reasons from an egocentric perspective, focusing on its immediate interactions with other agents. This requires the integration of multiple internal functions to perform a closed-loop operation and involves reciprocal inference and strategic negotiation in multi-agent interaction. Task completion relies primarily on the coordination between perception and control, allowing the system to react to other agents based on observed data without a deep understanding of complex social rules or underlying intent. Reasoning at this level manifests in several key strategies, including but not limited to the following:

- **Reactive Strategy (Cognition & Perception & Actuation).** A reactive strategy involves the system adjusting the state of the vehicle in direct response to real-time dynamic data from sensors. For instance, when the perception system detects that a preceding vehicle is decelerating, it autonomously engages the brakes to maintain a safe following distance and lane centering. Another example occurs when the system identifies a static obstacle on the road and rapidly plans and executes a local avoidance maneuver, which requires precise coordination of steering and speed.
- **Planning-based Strategy (Cognition & Multi-source Perception & Actuation).** A planning-based strategy involves the system executing a complete sequence of complex actions to achieve a clear objective within a specific scenario. This process depends on a comprehensive model of the environment constructed from fused sensor data. For example, during automated parking, the system integrates information from cameras and radar to understand the position and boundaries of a parking space. It then plans a coherent series of steering, forward, and reverse movements to guide the vehicle into the location, governed primarily by geometric and kinematic principles.

## 3.3 Social-Cognitive Level

This level encompasses tasks that are necessary for achieving full autonomy. It explicitly requires the system to emulate the advanced cognitive abilities of a human driver by incorporating an understanding of social commonsense, traffic regulations, and the predicted intentions of other agents. This constitutes a profound

level of social reasoning, demanding that the system operate as a socially-aware participant within the dynamic traffic environment. Key challenges at this level are illustrated by the following forms of reasoning:

- **Reasoning with Social Commonsense (Cognition & Perception & Actuation & Internal Knowledge).** This capability involves leveraging an internal world model to infer unstated context and anticipate likely outcomes. For instance, upon perceiving a ball rolling into the roadway, the system infers a high probability that a child is following. Consequently, the system preemptively reduces the vehicle speed. This response is not a simple perception-control reflex but an anticipatory judgment derived from a sophisticated world model.
- **Reasoning with Traffic Regulations (Cognition & Perception & Actuation & External Knowledge).** This form of reasoning requires the system to query and integrate an external knowledge base of traffic laws to guide planning and decision-making. For example, when navigating an intersection that lacks traffic signals, the system must retrieve and apply the relevant right-of-way regulation to determine the appropriate action. This involves applying a formal rule from an external source to a dynamic, real-world context.
- **Reasoning about Agent Intentions (Cognition & Perception & Actuation & Agent Modeling).**[1] This capability involves modeling the behavior of other intelligent agents to navigate complex, interactive scenarios. To merge into dense traffic, for example, the system must predict whether an adjacent vehicle will yield by observing the dynamics of that vehicle, such as changes in speed and trajectory. Based on this prediction, the system determines whether to accelerate for the merge or to slow down and await a new opportunity. Such interactions often require game-theoretic reasoning to model the reciprocal actions of intelligent agents.

### 3.4 From Cognitive Hierarchy to Core Challenges

Current AD systems, aligned with Society of Automotive Engineers (SAE) Levels 2 and 3 (Standard, 2021), have established foundational capabilities at both the Sensorimotor and Egocentric Reasoning levels. While proficient in basic perception-to-action loops and rule-based planning, their reasoning at the Egocentric level often lacks the flexibility to manage novel or highly interactive scenarios (Badue et al., 2021). The critical barrier to achieving higher automation (L4 and beyond), however, remains the largely unaddressed Social-Cognitive Level, which requires navigating complex, unwritten protocols of human social interaction in traffic.

The Survey Self-Driving Cars: A Survey (Badue et al., 2021) provides a comprehensive review of autonomy systems, showing that decision-making remains a central challenge, with perception no longer the dominant obstacle. Similarly, A Survey of Deep Learning Techniques for Autonomous Driving (Grigorescu et al., 2020) discusses how deep learning methods have successfully improved perception, yet still struggle with complex planning and behavior arbitration. A broader survey of surveys (Chen et al., 2023) further highlights that long-tail scenario reasoning and interaction still elude current architectures. Meanwhile, OccVLA (Liu et al., 2025b) augments perception with vision language reasoning to address semantic ambiguity in 3D scenes, implicitly acknowledging that perception alone is insufficient without higher-level inference. ORION (Fu et al., 2025) incorporates explicit reasoning mechanisms into trajectory prediction, motivated by the observation that failures often stem from incorrect logical interpretations of agent interactions rather than sensing errors. Similarly, Drive-R1 (Li et al., 2025a) enhances planning and decision-making through chain-of-thought-guided reinforcement learning, explicitly targeting reasoning deficiencies in long-horizon and safety-critical scenarios. The emergence of these surveys and methods across perception, prediction, and planning consistently reflects a shared premise: performance limitations in autonomous driving are increasingly dominated not by sensory accuracy, but by insufficient reasoning over complex, uncertain, and interactive environments.

Traditional methodologies, from intricate rule-based systems to purely data-driven models, are often too rigid to address the full spectrum of these higher-level reasoning challenges. Recent breakthroughs in LLMs and MLLMs, however, offer a more powerful and unified cognitive engine capable of enhancing Egocentric

---

[1]For such boundary tasks, it can serve either as an **Egocentric** prediction enhancement, or be elevated to a **Social-Cognitive** approach in more complex negotiation scenarios.

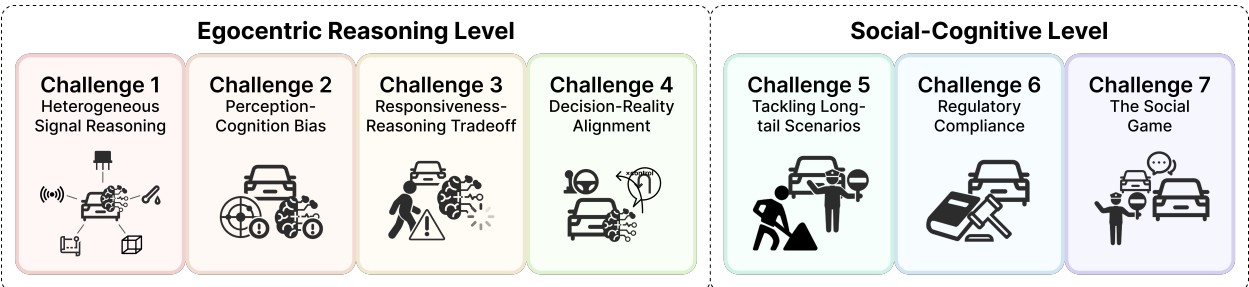

Figure 4: The taxonomy of seven core reasoning challenges in autonomous driving. These challenges are categorized by their corresponding cognitive level: C1–C4 at the Egocentric Reasoning level, while C5–C7 are at the Social-Cognitive level. Each numbered scenario illustrates a specific challenge analyzed in the text.

Reasoning while enabling Social-Cognitive capabilities (Hu et al., 2023). Their extensive world knowledge provides the context needed to anticipate agent intentions and understand implicit social norms. Furthermore, their sophisticated instruction-following abilities facilitate more robust planning in complex interactive scenarios and promote interpretable vehicle behavior. Critically, the architecture of these models is highly amenable to reinforcement learning, allowing for continuous adaptation in both direct agent interactions and broader social contexts (Ouyang et al., 2022). Collectively, these attributes position advanced reasoning models as a transformative technology for mastering the higher levels of the cognitive hierarchy.

However, adapting these general-purpose reasoning engines to autonomous driving systems remains a formidable challenge. A naive integration of such models is insufficient, necessitating a structured approach to identify and categorize the key difficulties. To establish this foundation, the following section introduces a comprehensive taxonomy of the reasoning challenges that must be addressed. This taxonomy serves as a framework to systematically analyze the limitations of current approaches and to guide future research toward the development of autonomous vehicles.

## 4 A Taxonomy of Reasoning Challenges in AD Systems

Achieving higher levels of vehicle autonomy requires mastering the complex reasoning demands of both the Egocentric Reasoning (C1–C4) and Social-Cognitive (C5–C7) levels. To provide a structured approach for tackling this multifaceted problem, this section introduces a taxonomy that deconstructs it into seven fundamental challenges. These challenges encompass a wide spectrum of functions that require higher-level cognition, from navigating direct agent interactions to understanding implicit social contexts. Each of these challenges is visually represented by the scenarios in Fig. 4 and is subsequently analyzed in detail.

### 4.1 C1 Challenge #1: Heterogeneous Signal Reasoning

An AD system must process and synthesize a multitude of heterogeneous data streams, including cameras, LiDAR, radar, and HD maps, to build a coherent representation of its environment. For example, a central reasoning engine must integrate these disparate signals, from raw sensor data to 3D point clouds. The core challenge lies in fusing this diverse information to support complex high-level reasoning.

This challenge manifests in several key areas of current model limitations. For instance, language models natively struggle with non-textual, numerical data like LiDAR point clouds, which necessitate robust mechanisms for cross-modal alignment. Furthermore, many contemporary vision models are fundamentally 2D-centric, creating a critical need for architectures that perform true 3D spatial reasoning across multiple views. The computational burden of processing high-resolution video streams also requires efficient information compression techniques that preserve temporal coherence. Finally, to effectively leverage the reasoning capabilities inherent to large language models, a shift is required from dense, unstructured feature

representations to object-level tokens that align the perceptual representation with the symbolic reasoning process (Hu et al., 2023).

These advanced reasoning capabilities are essential for critical driving tasks. For example, to understand complex spatial relationships, such as localizing an object "behind and to the left of a blue van" or determining if an approaching vehicle is facing forward, the system must move beyond simple 2D object detection. It must fuse images from multiple viewpoints with 3D geometric data to reason about the pose, orientation, and relative positioning of objects within a unified 3D space. Moreover, reasoning over high-resolution video streams is critical for tracking multiple objects and interpreting behavior over time, such as identifying a vehicle running a red light. This requires the system to dynamically associate visual cues with specific actors while maintaining temporal consistency.

### 4.2 `C2` Challenge #2: Perception-Cognition Bias

The reliability of an AD system is fundamentally challenged by uncertainty originating from both imperfect sensory input and fallible cognitive models. Extrinsic factors, such as adverse weather and sensor malfunctions, degrade perceptual data, while intrinsic factors, like model "hallucinations," introduce cognitive errors. Overcoming these challenges requires a robust reasoning mechanism capable of cross-modal validation, compensatory inference, and reality-checking to maintain a stable and accurate understanding of the environment.

Reasoning is essential for mitigating uncertainty from extrinsic sources. For example, adverse weather like rain and fog severely compromises optical sensors such as cameras and LiDAR. Simultaneously, the signals from radar, while capable of penetrating such conditions, are difficult to interpret due to complex reflective properties (Zhang et al., 2023b; Matos et al., 2024). A reasoning module must therefore dynamically weigh and fuse evidence from these disparate sources to form a coherent environmental representation. Similarly, when a primary sensor is occluded or fails, the system must employ compensatory reasoning. With sensor performance degraded, the system may infer the presence of a concealed hazard from a lead vehicle's sudden braking and decelerate preemptively, even without direct visual confirmation.

The system must also employ reasoning to counteract intrinsic cognitive failures, most notably model hallucinations. An MLLM might generate non-existent objects, such as a phantom traffic light on an open road, or omit critical, perceived objects like traffic cones from a descriptive inference. To address this, a reasoning layer must continuously perform reality-checking by cross-validating the outputs from the model against raw sensor data and map information. This process allows the system to filter fabricated entities, correct omissions, and align the final decision with the ground truth of the physical world.

### 4.3 `C3` Challenge #3: Responsiveness-Reasoning Tradeoff

AD systems face a fundamental trade-off between real-time responsiveness and the latency inherent to deep reasoning. In critical situations, a system must react within milliseconds. However, achieving this speed is complicated by three primary factors. First, the complicated reasoning processes of large models (LLMs and MLLMs) are computationally intensive and time-consuming (Tian et al., 2025a). Second, reliance on remote foundational models for inference introduces significant network latency, which is often incompatible with the stringent real-time demands of driving (Krentsel et al., 2024; Tahir & Parasuraman, 2025). Third, the massive data streams from high-resolution, multi-modal sensors impose a heavy computational load on onboard hardware, further hindering the efficient execution of sophisticated models (Wang et al., 2024c; Brown et al., 2023).

This tension between deliberative "slow thinking" and reactive "fast thinking" necessitates the development of dual-process architectures. Such a system must dynamically arbitrate between a fast, reactive controller for immediate responses and a deep, deliberative engine for complex planning and prediction (Serban et al., 2018). This challenge is particularly evident in scenarios that require the system to balance immediate action with strategic foresight. For example, a vehicle may need to execute an emergency maneuver to avoid a sudden obstacle, an action requiring the immediate activation of a reactive module. If this occurs during a complex, game-theoretic negotiation like a lane merge, the deliberative module must simultaneously evaluate

the long-term consequences associated with its actions, demanding seamless coordination between the two processes. This trade-off also manifests acutely during high-speed merging maneuvers. When entering dense highway traffic, the system must identify a safe gap in a fleeting time window, a task that requires both rapid perception of vehicle dynamics and deep reasoning about the intentions of other drivers. The final decision must synthesize these inputs with an assessment of the status of the ego-vehicle, where a slight delay risks a missed opportunity and a misjudgment in reasoning could lead to a collision.

### 4.4 C4 Challenge #4: Decision-Reality Alignment

A core challenge for AD systems is grounding high-level semantic decisions in the physical reality of the vehicle and its environment. A significant gap often exists between the abstract reasoning of large models and the concrete requirements of motion planning, leading to inconsistencies between a conceptual decision and the final executable trajectory (Yao et al., 2024). For instance, a chain-of-thought output like "change to the right lane to avoid the obstacle" may be conceptually sound but physically infeasible if it disregards the kinodynamic constraints of the vehicle (e.g., turning radius, tire adhesion) or the geometric constraints of the environment. This misalignment can produce plans that are unsafe or impossible to execute. Therefore, a reasoning mechanism is imperative to ensure that high-level strategic decisions are continuously validated against real-world physical laws and constraints.

The necessity of this reasoning is evident in common yet complicated scenarios. For example, when maneuvering on a narrow road, a high-level decision to "reverse to yield" is insufficient. The reasoning module must further assess the physical viability of this plan by evaluating the clearance behind the vehicle and ensuring the intended trajectory does not conflict with roadside obstacles. This challenge is amplified in dynamic and uncertain conditions, such as navigating a sharp turn on a slippery surface. A high-level plan for a sharp turn may be invalidated by the physical reality of reduced tire adhesion. Reasoning must connect the semantic goal of "navigate the turn" to the physical state of the environment, inferring a drastically reduced coefficient of friction for the road surface. This inference is then used to compute a safe vehicle speed, preventing a loss of control that could occur if the decision were based solely on the posted speed limit.

### 4.5 C5 Challenge #5: Tackling Long-tail Scenarios

The long-tail distribution of driving scenarios poses a fundamental challenge to prevailing data-driven methodologies, which falter in novel or data-scarce situations. While current models perform well in common scenarios, their efficacy sharply degrades when encountering rare events like temporary construction zones or extreme weather (Bogdoll et al., 2022; Tian et al., 2024b). This problem stems from two interconnected limitations: the infeasibility of comprehensive data collection and the inherent brittleness of pattern-matching mechanisms. Consequently, the key to handling long-tail events lies not in accumulating more data, but in equipping the system with robust reasoning capabilities to make sound decisions without direct experiential precedent.

The first limitation is data scarcity. Long-tail events are, by definition, rare, making the acquisition of sufficient real-world training data logistically impossible (Zhang et al., 2023a). While data synthesis offers a partial remedy, generating high-fidelity, physically realistic simulation data remains a formidable challenge, and it is impossible to enumerate all potential corner cases beforehand (Dosovitskiy et al., 2017; Wang et al., 2021). The existence of these "unknown unknowns" places a ceiling on any strategy that relies on exhaustive data coverage. Therefore, an autonomous system must be able to generalize from underlying commonsense principles rather than merely interpolating from seen data. The second limitation is the operational mechanism of current models, which relies on pattern matching rather than logical inference. This approach is effective at generalizing within the training distribution but becomes brittle when confronted with out-of-distribution long-tail events (Grigorescu et al., 2020). Such scenarios often involve dynamic rule changes and conflicting information, which demand causal and logical deliberation that pattern matching cannot provide.

The critical role of reasoning is evident in how a system handles such scenarios. For instance, when a pedestrian suddenly emerges from an occluded area, a system limited to pattern matching may fail without

a direct precedent. In contrast, a reasoning-enabled system can proactively apply defensive driving principles, using commonsense knowledge that occlusions create blind spots or inferring that a wrong-way vehicle signals a downstream anomaly to reduce speed. This inferential capability is also essential for resolving rule conflicts. As an example, in the temporary construction zone, hand signals from a traffic officer instruct the vehicle to stop, contradicting a green traffic light. The challenge here is not perceptual but decision-theoretic. The system must apply a known hierarchy of authority (e.g., human officer $\succ$ traffic signal $\succ$ map data) to deduce that standing regulations are temporarily superseded and execute the action corresponding to the highest-priority command.

### 4.6  C6 Challenge #6: Regulatory Compliance

An AD system must navigate a complicated and dynamic web of traffic regulations to ensure safe and legal operation. This requirement presents a significant reasoning challenge, which can be deconstructed into two primary dimensions: the complexity of interpretation and context-dependence. The difficulty in interpretation stems from a regulatory landscape that includes not only thousands of legal statutes but also numerous local ordinances and unwritten driving conventions (Koopman & Wagner, 2017; Lin, 2015). A scenario in which a vehicle must reason about legal statutes, local customs, and regional standards to determine proper bus-lane use highlights this complexity. Compounding this issue, traffic laws vary significantly across different jurisdictions, necessitating robust system capabilities for retrieving and interpreting applicable rules in any given context (Dixit et al., 2016). Furthermore, the application of these rules is highly dependent on the immediate situation. The system must continuously analyze the behavior of other road users, current road conditions, and environmental factors to select and prioritize relevant regulations.

Addressing this challenge demands a multi-step reasoning process: from scene perception, the system must retrieve relevant regulations, judge their applicability, and resolve any conflicts among them to yield a safe and lawful decision. Numerous driving scenarios illustrate this requirement. For instance, consider unprotected turns at intersections, which research indicates account for a substantial portion of traffic accidents (Li et al., 2019). In these settings, the system must accurately infer the assignment of right-of-way and integrate this inference with predictions regarding the intent of other vehicles and a comprehensive risk assessment. Another critical scenario involves the system's response to emergency vehicles or temporary traffic controllers. The detection of such an event must trigger a specialized reasoning protocol to retrieve and execute overriding rules, such as yielding to an ambulance, even if this action requires temporarily violating standard traffic laws. This highlights the need for system-level, robust, dynamic prioritization among conflicting regulations.

### 4.7  C7 Challenge #7: The Social Game

In mixed-traffic environments, an autonomous vehicle must operate not merely as a law-abiding agent but as a socially intelligent participant. This requires the system to reason about the implicit, non-verbal communication of other road users to infer their intentions and anticipate their actions. Existing models, however, often treat human drivers as passive agents, failing to interpret the subtle modulations in speed and headway used to signal intent. This can lead to interactions that are overly conservative, inefficient, or dangerously aggressive (Li et al., 2024a; Poots, 2024).

Beyond inferring the intent of others, the system's own actions must be legible and its decisions transparent. The absence of human-like cues can make the behavior of the vehicle unpredictable to pedestrians, creating unsafe and uncomfortable situations. Furthermore, the opaque nature of end-to-end models undermines the trust from passengers and regulators, as the rationale behind decisions is not accessible (Rezwana & Lownes, 2024; Zhanguzhinova et al., 2023). Therefore, a reasoning-enabled system must not only interpret social cues but also generate behavior that is socially compliant and readily understandable.

The application of social reasoning is critical across three primary interaction domains. First, in vehicle-to-vehicle interactions like merging, the system must decode and respond to subtle cues. Human drivers use changes in speed and headway to implicitly signal the intent to yield or proceed, and the autonomous system must participate in this dynamic negotiation (Naiseh et al., 2025; Nozari et al., 2024). Second, in vehicle-to-pedestrian scenarios, such as the unsignalized crosswalk, the system must infer the intent to cross from posture and movement. It must then project its own yielding intent through clear and considerate actions,

such as early and smooth deceleration, to establish a shared understanding. Finally, for vehicle-to-passenger interaction, the system must provide transparent explanations for its actions. The ability to articulate the rationale for a sudden maneuver in natural language linking sensor inputs to a high-level decision is crucial for building trust with the occupants of the vehicle.

***Summary & Discussion.*** The preceding analysis of seven core challenges collectively underscores a necessary paradigm shift in the development of autonomous systems: a move from pure perception and control to sophisticated, human-like reasoning. To operate safely and effectively in a complex world, an autonomous system must master a spectrum of reasoning capabilities. The essential challenges that we have detailed are summarized as follows:

Egocentric Reasoning Level (several challenges remain unresolved):

- **Heterogeneous Signal Reasoning:** Systems must fuse disparate data types to build a coherent and unified world model as the prerequisite for all subsequent reasoning.
- **Perception-Cognition Bias:** Reasoning is required to validate information and compensate for failures arising from environmental factors or intrinsic model limitations, ensuring the world model is reliable.
- **Responsiveness-Reasoning Tradeoff:** The high latency inherent to large models must be reconciled with the millisecond-level reaction times required for driving, pointing toward hybrid architectures that balance deliberation and reaction.
- **Decision-Reality Alignment:** High-level semantic decisions must be continuously aligned with vehicle kinodynamic constraints and environmental physical laws to ensure that all plans are executable.

Social-Cognitive Level (Largely Unsolved):

- **Tackling Long-tail Scenarios:** In novel edge cases where direct experience is absent, systems must rely on commonsense and social context to navigate situations that defy standard patterns.
- **Regulatory Compliance:** Systems must interpret and adhere to the complex and dynamic set of formal societal rules embodied in traffic laws to ensure all actions are legally compliant.
- **The Social Game:** The system must infer the implicit, informal rules of human interaction, interpreting the intent of other agents and engaging in legible, socially acceptable negotiation.

This comprehensive analysis reveals a critical insight. While the initial set of challenges represents profound engineering hurdles in building and grounding a reliable agent, the ultimate bottleneck to achieving human-like autonomy lies in navigating the complexities of the social world. Addressing these latter challenges requires a fundamental move beyond pattern matching. It demands the deep, structured reasoning capabilities promised by large models. The pivotal open question for future research is therefore how to integrate these powerful reasoning engines into AD systems, guaranteeing the absolute reliability and real-time performance required for deployment in the physical world.

# 5 System-Centric Approaches: Architectures and Methodologies

## 5.1 Bridging the Paradigm Gap: From General Reasoning to Situated Autonomy

A fundamental paradigm gap separates the reasoning required for AD from the general reasoning capabilities of LLMs and MLLMs. This disparity manifests across several critical dimensions, most notably in their core objectives, operational domains, and input modalities.

**Goal & Determinism.** The primary mandate of an AD system is to ensure safety and control within the physical world. This mandate requires a reasoning framework that is highly deterministic and predictable. Given identical perceptual input, the decisions of the system must remain stable and consistent, as its logic is grounded in rigorous physical laws, traffic regulations, and control theory. Conversely, the core objective of an LLM or MLLM is to operate within the informational domain. The reasoning process is inherently probabilistic and creative, prioritizing semantic coherence and plausibility over physical precision. As a result, an LLM or MLLM can generate multiple distinct yet valid responses to a single prompt.

**Input & Environment.** The input to autonomous driving consists of real-time, high-dimensional, and continuous physical world data from sensors (e.g., cameras, LiDAR, and millimeter wave radar). Real physical environments are full of noise and uncertainty and must be processed in real time. In contrast, the input to LLMs mainly comprises discrete symbolic textual data. LLMs operate within a relatively closed virtual world constructed by linguistic symbols. Even when external information is acquired through tool calling, their interaction mode remains structured.

In summary, reasoning in autonomous driving is a deterministic, safety-critical process engineered for precise action in the physical world. The reasoning of LLMs and MLLMs, conversely, is a probabilistic, generative process designed for semantic flexibility in the informational world. This fundamental paradigm gap necessitates innovative approaches to adapt and ground the powerful general reasoning within the specialized context of autonomous systems. The following taxonomy provides a systematic review of these emerging system-centric innovations.

## 5.2 A Task-Based Taxonomy within System-Centric Approaches

A significant body of research now focuses on bridging the paradigm gap by adapting foundation models to the specific demands of autonomous driving. These efforts universally leverage techniques such as Chain-of-Thought (CoT) reasoning and structured representations to endow systems with more robust capabilities. To provide a clear narrative, we structure our taxonomy to reflect a logical progression from enhancing individual components to building holistic intelligent agents. As depicted in Fig. 5, the timeline presents the chronological release of each major method, illustrating the rapid pace at which this domain is evolving. Our taxonomy groups the research landscape into thematically ordered stages of rising complexity.

Our taxonomy commences with a review of methods that enhance the foundational modules for environmental understanding (Fig. 6): perception (Sec. 5.2.1) and prediction (Sec. 5.2.2). It then transitions to the core action-generation module responsible for creating safe and coherent behaviors: planning and decision-making (Sec. 5.2.3). Subsequently, we explore holistic architectures that transcend the traditional modular pipeline by creating more integrated and end-to-end agents (Sec. 5.2.4). Our review culminates with an examination of supporting and auxiliary tasks that enable robust system development and validation (Sec. 5.2.5). Throughout this taxonomy, we systematically connect the discussed approaches to the core challenges (C1–C7) identified in Sec. 4. This genealogical view, complemented by a comprehensive comparison table (see Table 1 in Appendix B), serves as a quick reference for researchers to navigate the relationships and core attributes of existing methods.

### 5.2.1 Reasoning-Enhanced Perception

Recent research in perception primarily focuses on moving beyond simple object detection towards a deeper, contextual understanding of the driving scene. The goal is to enhance the visual comprehension, robustness, and structured representation capabilities of MLLMs.

Key works in this area exemplify several strategic directions. One direction involves architectural innovations to improve fine-grained perception. For instance, as shown in Fig. 6 (a), HiLM-D (Ding et al., 2025a) employs a two-stream framework to better interpret and perceive high-risk objects, addressing how standard models often fail to capture critical multi-scale details (C1, C2). DMAD (Shen et al., 2025) proposes explicit decoupling and restructuring of motion and semantic representations, and mitigates heterogeneous signal interference via architecture-level reasoning, thereby enhancing autonomous driving perception (C1). OccVLA (Liu et al., 2025b) proposes a novel occupancy-visual-language framework, which achieves more critical fine-grained understanding of 3D spatial semantics and addresses the inference latency issue caused by the massive number of parameters in autonomous driving models based on 3D vision-language models (C1, C3). A second strategy focuses on data-centric contributions to enable robust evaluation. Reason2Drive (Nie et al., 2024) contributes a large-scale video-text dataset designed specifically for the structured assessment of perception and reasoning. This work directly addresses data scarcity and provides a new benchmark for verifying genuine scene comprehension (C1, C5). A third direction centers on creating an explicit link between visual evidence and language-based reasoning. ARGUS (Man et al., 2025) utilizes bounding boxes

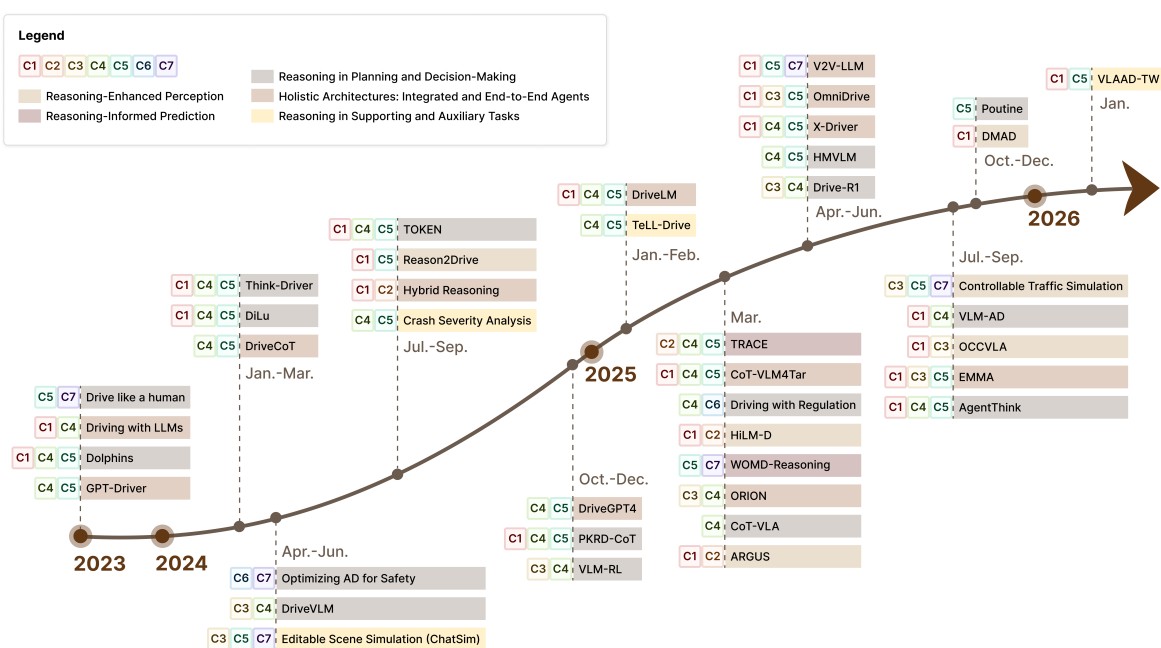

Figure 5: The chronological evolution of major methods in autonomous driving. This timeline highlights the rapid progression of the field and provides historical context for the thematic complexity taxonomy introduced in Sec. 5.2.

as explicit signals to guide the visual attention of the model, improving performance in tasks that require precise visual grounding (C1, C2).

### 5.2.2 Reasoning-Informed Prediction

In the context of AD, prediction involves forecasting the future behavior of other traffic participants. Current research in this area increasingly focuses on enhancing robustness in long-tail scenarios, augmenting data with domain knowledge, and ensuring that predictions align with safe decision-making.

Several distinct research strategies highlight this trend. One strategy focuses on improving predictive robustness in situations with sparse or uncertain observations. For example, as shown in Fig. 6 (b), TRACE (Puthumanaillam et al., 2025) improves behavior prediction by using Tree-of-Thought (ToT) and counterfactual criticism, which allows the model to generate and evaluate multiple hypothetical reasoning paths (C2, C4, C5). Another strategy involves injecting explicit domain knowledge into the models. WOMD-Reasoning (Li et al., 2024c) accomplishes this by generating millions of question-answer pairs based on traffic rules, thereby enriching the data resources for both prediction and decision-making (C5, C7). A third direction seeks to ensure consistency between prediction and subsequent planning.

### 5.2.3 Reasoning in Planning and Decision-Making

This category reviews works that focus specifically on the core action-generation module of the autonomous driving stack, which is responsible for planning the ego-vehicle's trajectory and making tactical decisions. Research in this domain has evolved significantly beyond traditional trajectory optimization. The central goal is to create a reasoning process that is not just optimal, but also compliant with external rules, coherent in its internal logic, and adaptive to an open-world environment. We organize our review around these three central themes.

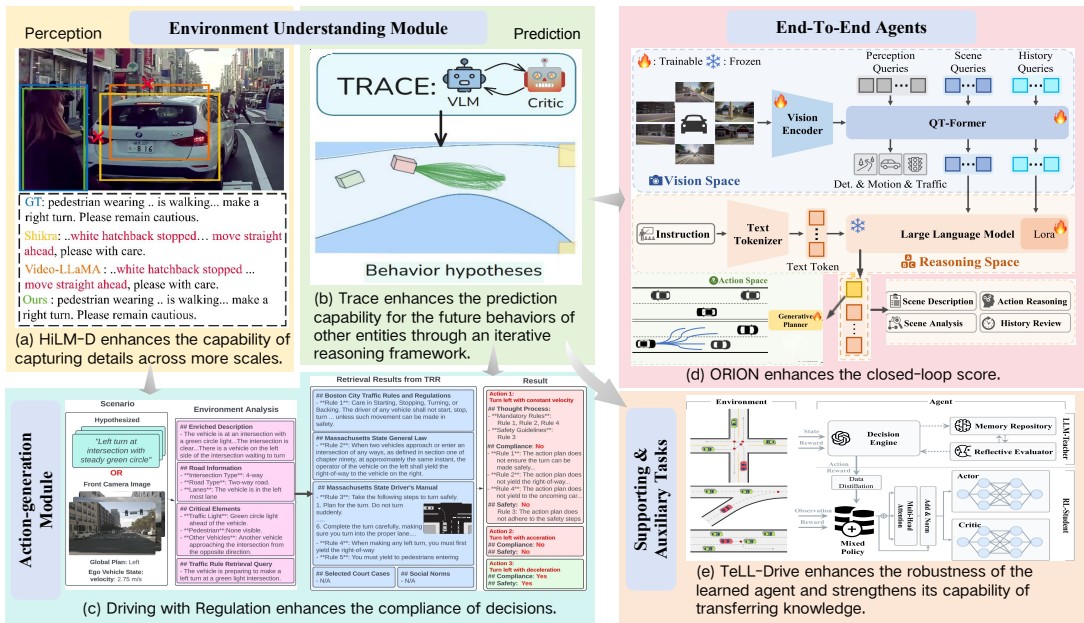

Figure 6: Among various existing methods, we select and present five cases that enhance capabilities across different dimensions in an AD system: (a) HiLM-D (Ding et al., 2025a) focuses on Reasoning-Enhanced Perception; (b) Trace (Puthumanaillam et al., 2025) focuses on Reasoning-Informed Prediction; (c) Driving with Regulation (Cai et al., 2024) focuses on Reasoning in Planning and Decision-Making; (d) ORION (Fu et al., 2025) focuses on Holistic Architectures; (e) Tell-Drive (Xu et al., 2025a) focuses on Reasoning in Supporting and Auxiliary Tasks.

**Embedding External Constraints for Compliant Behavior.** A primary research thrust is the explicit integration of external rules and safety norms into the reasoning framework. This marks a paradigm shift from optimizing for pure performance to optimizing for compliance and social acceptability. These approaches ensure that an agent's behavior is grounded in the complex realities of the road. For instance, some methods directly tackle legal and safety verification. As shown in Fig. 6 (c), Driving with Regulation (Cai et al., 2024) integrates a regulation retrieval mechanism with CoT reasoning, enabling the system to generate decisions that are certifiably compliant with local traffic laws (C4, C6). Optimizing AD for Safety (Sun et al., 2024) combines RLHF with LLMs, using physical and physiological feedback to train an agent that prioritizes human safety (C6, C7). Poutine (Rowe et al., 2025) utilizes vision-language-trajectory pre-training and preference-based reinforcement learning to align the model's predictions with established driving preferences and safety regulations (C2, C5). Extending beyond hard constraints, other works focus on social compliance. Drive Like a Human (Fu et al., 2024) emphasizes that systems should exhibit human-like driving styles and be able to interact naturally with their environment, addressing the crucial challenges of social interaction and long-tail events (C5, C7).

**Ensuring Internal Coherence through Structured Reasoning.** Another body of research focuses on ensuring consistency between the model's high-level semantic reasoning and its final low-level executable plan. This addresses the critical challenge of reasoning-action alignment. Chain-of-Thought has become a cornerstone technique for achieving this coherence by making the "thought process" explicit. Works like CoT-VLA (Zhao et al., 2025), Think-Driver (Zhang et al., 2024b), and PKRD-CoT (Luo et al., 2024) incorporate explicit reasoning chains to generate transparent and justifiable driving instructions, directly improving interpretability and the alignment between reasoning and action (C1, C4, C5). To further ground this reasoning process, many methods utilize structured representations. For example, TOKEN (Tian et al., 2024a) decomposes scenes into object-level tokens, providing a scaffold that helps the LLM apply its common-sense knowledge to long-tail planning scenarios (C1, C4, C5). HMVLM (Wang et al., 2025a) employs a multi-stage CoT process that moves from scene understanding to decision-making to trajectory inference, ensuring

a logical flow (C4, C5). A variety of VLM-based architectures also contribute to this theme. For instance, VLM-AD (Xu et al., 2024a) uses a VLM as a teacher to provide rich reasoning labels, while DriveVLM (Tian et al., 2024c) and Dolphins (Ma et al., 2024a) use hybrid systems and multimodal CoT to achieve human-like adaptability. Finally, reinforcement learning (RL) is increasingly used to fine-tune this alignment (C1, C3, C4, C5). VLM-RL (Huang et al., 2024) uses a pre-trained VLM to generate dense reward signals, while Drive-R1 (Li et al., 2025a) connects CoT reasoning to the RL policy to improve decision quality (C3, C4).

**Achieving Open-World Adaptability through Learning and Interaction.** Furthermore, recent works are advancing planning and decision-making models from closed systems to open agents that can learn from experience and interact with external knowledge. This pushes the boundaries of adaptability, especially for handling novel situations. DiLu (Wen et al., 2024) pioneers this direction by enabling a model to accumulate experience and correct past mistakes through internal memory and reflection mechanisms (C1, C4, C5). This transforms decision-making from a series of independent, one-time inferences into a dynamic process of continuous learning. Pushing this concept further, AgentThink (Qian et al., 2025) trains an agent to dynamically call external tools (e.g., calculators, search engines). This significantly enhances the depth and flexibility of its reasoning process, allowing it to solve complex problems that require external knowledge or precise computation, thereby evolving the agent from a "closed-world planner" to an "open-world problem solver" (C1, C4, C5).

### 5.2.4 Holistic Architectures: Integrated and End-to-End Agents

This category reviews works that move beyond enhancing individual modules to architecting more holistic intelligent agents. The primary motivation is to overcome the information bottlenecks and error propagation inherent in traditional sequential pipelines. Research in this area follows a trajectory that begins with the tight coupling of key modules, progresses toward fully unified end-to-end reasoning systems, and finally expands the scope of what a holistic agent can do.

**Integrating Perception and Decision-Making.** The first step toward holistic design involves creating architectures that directly integrate perception and decision-making modules. These works reframe visual reasoning not as a passive representation learning task but as an active, decision-oriented explanatory process. For instance, some research focuses on breaking through visual reasoning bottlenecks in complex scenarios. DriveLM (Sima et al., 2024) uses a graph-structured VQA dataset to handle complex scene interactions, while CoT-VLM4Tar (Ren et al., 2025) employs CoT reasoning to resolve abnormal traffic situations like phantom jams (C1, C4, C5). Other works improve the grounding of reasoning through structured multimodal inputs. Driving with LLMs (Chen et al., 2024) achieves this by fusing object geometric vectors with language, while Hybrid Reasoning (Azarafza et al., 2024) integrates mathematical and commonsense reasoning to improve robustness in adverse weather (C1, C2, C4). Collectively, these approaches aim to improve closed-loop performance and interpretability, as exemplified by X-Driver (Liu et al., 2025c), which uses an autoregressive model to generate decision commands from vision-language inputs (C1, C4, C5).

**The Evolution to End-to-End Reasoning Agents.** The ultimate goal of this research direction is the development of fully end-to-end agents that map sensor inputs directly to control outputs. A critical evolution in this domain is the shift from opaque "black-box" models to interpretable "glass-box" systems. The foundation for this shift is often laid by new datasets, such as DriveCoT (Wang et al., 2024b), which provide end-to-end data with explicit CoT labels (C4, C5). Building on this, architectures like GPT-Driver (Mao et al., 2023) and DriveGPT4 (Xu et al., 2024b) have pioneered the glass-box concept by generating explicit natural language justifications alongside vehicle actions (C4, C5). These models must also handle diverse inputs. EMMA (Hwang et al., 2025), for instance, fuses non-perceptual inputs like navigation instructions with visual data, though this highlights the trade-off between reasoning depth and real-time performance (C1, C3, C5). Perhaps the most critical technical challenge is ensuring consistency between high-level semantic reasoning and low-level numerical trajectories. As shown in Fig. 6 (d), ORION (Fu et al., 2025) directly tackles this semantic-numerical alignment gap with a generative planner, improving closed-loop scores and real-world applicability (C3, C4).

**Expanding the Scope of Holistic Reasoning.** Beyond single-agent architectures, cutting-edge research is expanding the scope of reasoning to handle more complex interactions and scenarios. One major frontier is shifting the focus from performance in average scenarios to robust reasoning in extreme conditions. OmniDrive (Wang et al., 2024a) exemplifies this by using a dataset built on counterfactual reasoning to evaluate decision-making in rare, long-tail events, pushing the field to address the most difficult cases (C1, C3, C5). Another frontier involves expanding from single-vehicle intelligence to collective multi-agent intelligence. V2V-LLM (Chiu et al., 2025) introduces a vehicle-to-vehicle question-answering framework that enables inter-vehicle perceptual fusion and cooperative planning. This represents a critical step towards addressing complex social interaction challenges and realizing the potential of swarm intelligence on the road (C1, C5, C7).

### 5.2.5 Reasoning in Supporting and Auxiliary Tasks

Beyond the core driving pipeline, reasoning is also being applied to crucial supporting tasks that form the development and validation ecosystem. These efforts are critical for creating robust and scalable solutions. The research in this area can be broadly categorized into innovations in learning paradigms and the development of reasoning-driven tools for data generation and analysis.

**Learning Paradigms.** Some research focuses on improving the training process itself, aiming to make learning more efficient and generalizable. As shown in Fig. 6 (e), TeLL-Drive (Xu et al., 2025a) introduces a teacher-student framework where a powerful teacher LLM guides a more compact student deep reinforcement learning (DRL) policy. This approach improves the robustness of the learned agent and enhances its ability to transfer knowledge across different driving conditions. Such innovations address the core challenges of reasoning-decision alignment (C4) and long-tail generalization (C5) by focusing on the scalability of the training paradigm itself, rather than just the model architecture.

**Reasoning-Driven Simulation and Analysis.** A major focus in auxiliary tasks is the generation and analysis of high-fidelity, diverse, and controllable scenarios, which are essential for safely testing agents in rare or dangerous situations. Several works leverage the generative capabilities of LLMs to create rich simulation environments. For instance, Editable Scene Simulation (ChatSim) (Wei et al., 2024c) allows developers to use natural language to edit and compose realistic 3D scenes for simulation training (C3, C5, C7). Similarly, Controllable Traffic Simulation (Liu et al., 2024c) uses hierarchical CoT reasoning to generate complex and controllable traffic scenarios (C3, C5, C7). In a related direction, other tools use reasoning for offline data analysis. Crash Severity Analysis (Zhen et al., 2024) applies CoT reasoning to assess traffic accident reports, providing deeper, interpretable insights into event causation and severity (C4, C5). VLAAD-TW (Kondapally et al., 2026) introduces a spatiotemporal vision-language reasoning framework and dataset to enhance autonomous driving understanding under transitional weather, serving as auxiliary reasoning support beyond core perception/prediction modules.

## 5.3 Discussion and Insights: Prevailing Trends

**Prevailing Trends.** This review of system-centric approaches reveals several dominant trends in the effort to integrate reasoning into autonomous systems:

- **Holistic Architectures:** A clear trajectory exists away from optimizing isolated modules and toward designing holistic, reasoning-centric architectures. Techniques like CoT are applied across the entire stack to serve as a cognitive backbone.
- **Internal Coherence:** A strong focus is placed on ensuring internal coherence. CoT and structured representations are widely adopted to bridge the gap between high-level semantic decisions and low-level executable actions, a direct response to the Decision-Reality Alignment (C4) challenge.
- **Interpretability:** An architectural shift is underway from opaque "black-box" models to interpretable "glass-box" systems. These systems provide explicit justifications for their actions, addressing demands for transparency in the Social Game (C7).

- **Open-World Adaptability:** An emerging trend targets open-world adaptability. Methods that incorporate memory (e.g., DiLu) or external tools (e.g., AgentThink) signify a move from "closed-world planners" to "open-world problem solvers" capable of handling novel long-tail scenarios (C5).
- **Compliance over Optimization:** A significant paradigm shift involves prioritizing compliance over pure optimization. A growing body of work explicitly embeds external constraints, such as traffic laws (C6) and social norms (C7), directly into the decision-making framework.

**Cognitive Hierarchy-Informed Design Principles.** Viewed through the cognitive hierarchy, recent methodological advances can be interpreted as strengthening differentiated reasoning layers. At the Ego-centric Level, works such as HiLM-D (Ding et al., 2025a), TOKEN (Tian et al., 2024a), HMVLM (Wang et al., 2025a), and ORION (Fu et al., 2025) enhance internal coherence and decision-reality alignment through object-centric representations, structured reasoning, multi-stage inference, and semantic-trajectory alignment. At the Social-Cognitive Level, approaches including Driving with Regulation (Cai et al., 2024), Optimizing AD for Safety (Sun et al., 2024), and Drive Like a Human (Fu et al., 2024) incorporate rule grounding, risk awareness, and socially consistent interaction. Together, these developments shift the focus from isolated perception improvements toward structured reasoning across cognitive levels.

From the perspectives of training and system design, cognitive theory suggests three principles. First, hierarchical capability allocation: low-level sensing and control modules maintain determinism and physical verifiability, while high-level modules handle rule retrieval, intention modeling, and conflict arbitration, improving system stability and controllability. Second, risk-sensitive objective design: asymmetric penalties for high-risk errors align training objectives with societal safety expectations rather than solely trajectory accuracy. Third, dual-process architectures: combining fast reactive modules with trigger-based deep reasoning modules enables high-level inference only under uncertainty or rule conflicts, balancing real-time constraints and decision quality.

In conclusion, the integration of reasoning has injected a powerful new paradigm into autonomous driving research, opening promising avenues for solving long-standing challenges in long-tail scenarios and social interaction. However, the true viability of these system-centric innovations hinges on the ability to rigorously and reliably measure their performance. The development of such evaluation methods is therefore a critical and parallel research frontier, which we explore in the following section.

## 6 Evaluation-Centric Practices: Benchmarks and Datasets

Complementing the system-centric innovations discussed previously, this section examines the parallel and equally critical frontier of evaluation. Progress in developing robust reasoning agents is intrinsically tied to the methodologies used to validate their performance. In the domain of autonomous driving reasoning, the construction of benchmarks and datasets is not merely a supporting activity; it constitutes a critical research area that delineates the boundaries for model learning and ensures rigorous scientific validation. This section provides a systematic review of these evaluation-centric practices, organized by the increasing complexity of the capabilities they are designed to measure.

### 6.1 A Thematic Taxonomy of Evaluation Practices

Traditional evaluation metrics for AD, such as collision rates or trajectory error, are adept at quantifying physical performance but are insufficient to assess the cognitive capabilities of a reasoning-based agent. These metrics can confirm what a system does, but often fail to reveal why it makes a certain decision or whether its underlying reasoning is sound. To address this evaluation gap, a new generation of benchmarks and datasets is emerging that prioritizes the measurement of cognitive skills over purely physical outcomes.

This section organizes a review of these evaluation-centric practices. To provide historical context for this review, Fig. 7 chronologically marks the release of each major benchmark, offering a visualization of the rapid evolution of this research domain. Our taxonomy organizes the landscape into a thematic progression of increasing complexity. We begin with benchmarks designed to evaluate foundational cognitive abilities (Sec. 6.1.1). We then examine platforms that assess system-level and interactive performance in dynamic

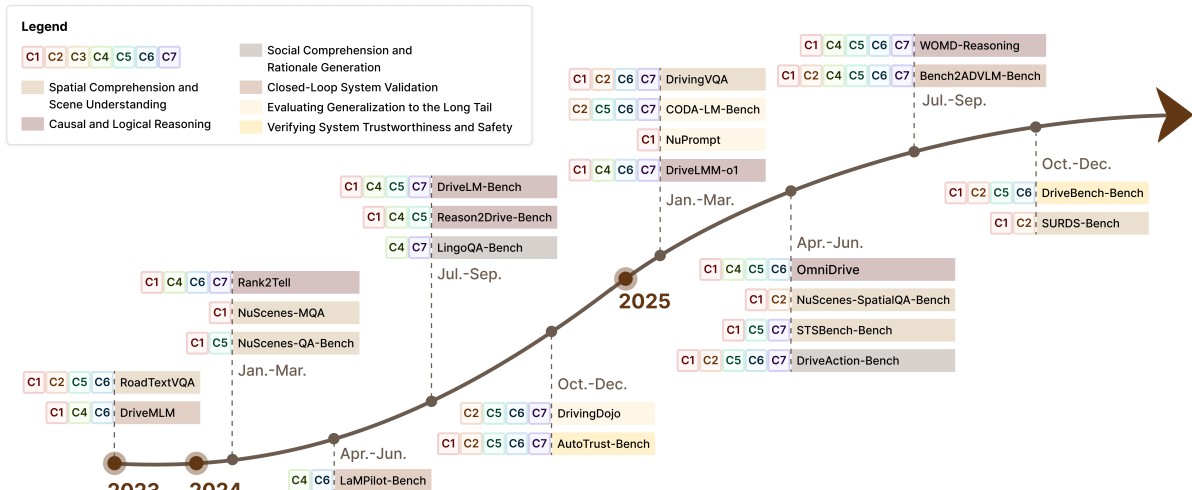

Figure 7: The chronological evolution of benchmarks and datasets for autonomous driving reasoning. This timeline illustrates the rapid acceleration of evaluation-centric research and provides the historical context for the thematic taxonomy discussed in Sec. 6.1.

closed-loop environments (Sec. 6.1.2). Finally, we review benchmarks that focus on robustness and trustworthiness, specifically by stress-testing agents in long-tail and safety-critical scenarios (Sec. 6.1.3). Throughout this taxonomy, we systematically connect the discussed benchmarks and datasets to the core challenges (C1–C7) identified in Sec. 4.

### 6.1.1 Evaluating Foundational Cognitive Abilities

The ability to perform complex actions in the real world is predicated on a set of foundational cognitive skills. We review benchmarks designed to measure these core competencies, the essential precursors to intelligent driving behavior. These evaluations typically operate in an open-loop question-answering format, focusing on what the agent understands about a static or prerecorded scene rather than how it acts within it.

**Spatial Comprehension and Scene Understanding.** This category of benchmarks evaluates an agent's ability to accurately perceive and model three-dimensional spatial relationships, a fundamental requirement for any physical agent. As a pioneering work, NuScenes-QA-Bench (Qian et al., 2024) established the first large-scale VQA benchmark for autonomous driving as shown in Fig. 8 (a) (C1, C5). This foundation has inspired a family of more advanced evaluations. For instance, NuScenes-MQA (Inoue et al., 2024) scales the volume of questions (C1), while NuScenes-SpatialQA-Bench (Tian et al., 2025b) focuses specifically on complex spatial reasoning by generating answers directly from ground-truth LiDAR data, thereby mitigating the impact of perception model errors (C1, C2). To further refine this evaluation, SURDS-Bench (Guo et al., 2025) systematically assesses fine-grained spatial capabilities across distinct categories such as orientation and depth (C1, C2). This area is also complemented by other specialized benchmarks, including RoadTextVQA (Tom et al., 2023) for scene text comprehension (C1, C2, C5, C6) and STSBench-Bench (Fruhwirth-Reisinger et al., 2025) for evaluating spatiotemporal reasoning in complex multi-agent interactions (C1, C5, C7).

**Causal and Logical Reasoning.** Moving beyond spatial awareness ("what" and "where"), this class of benchmarks assesses an agent's ability to perform multi-step and logical inference to understand "how" and "why" events unfold. The primary methodology here is evaluating explicit reasoning chains. For example, DriveLM-Bench (Sima et al., 2024) adopts a graph-structured VQA framework that spans the entire pipeline from object detection to planning decisions (C1, C4, C6). Similarly, Reason2Drive-Bench (Nie et al., 2024) provides over $600,000$ video-text pairs that explicitly decompose the driving process into perception,

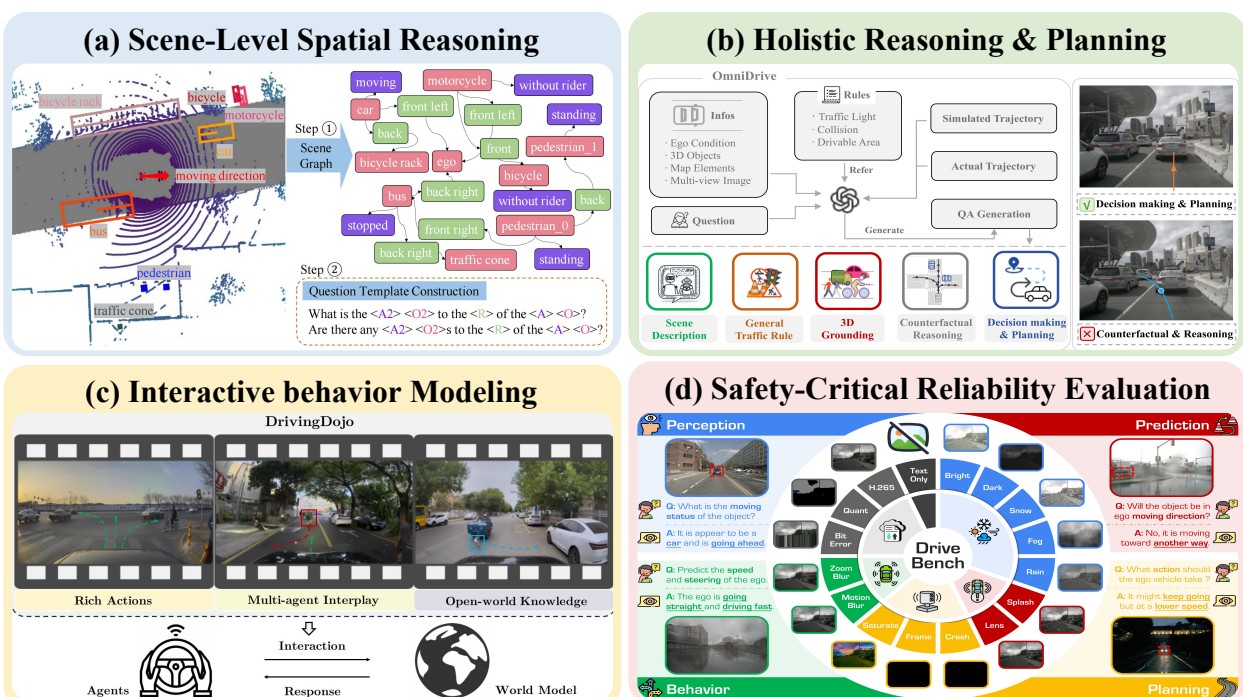

Figure 8: Among various existing benchmarks, we highlight four widely used cases: (a) NuScenes-QA for scene-level spatial reasoning (Qian et al., 2024); (b) OmniDrive for holistic reasoning and planning (Wang et al., 2025b); (c) DrivingDojo for interactive behavior modeling (Wang et al., 2024e); (d) DriveBench for safety-critical reliability evaluation (Xie et al., 2025).

prediction, and reasoning steps (C1, C4, C5). Other datasets, like DriveLMM-o1 (Ishaq et al., 2025) (C1, C4, C6, C7) and WOMD-Reasoning (Li et al., 2024b) (C1, C4, C5, C6, C7), provide millions of annotated question-answer pairs focused on reasoning about interactions governed by traffic rules and human intentions. DrivingVQA (Corbière et al., 2025) specifically tests rule-based logical reasoning by deriving questions from official driving theory exams (C1, C2, C6, C7). More advanced forms of reasoning are also targeted. As shown in Fig. 8 (b), OmniDrive (Wang et al., 2025b) employs counterfactual reasoning to link plans with explanations (C1, C4, C5, C6), while Rank2Tell (Sachdeva et al., 2024) is designed for multimodal importance ranking and rationale generation (C1, C4, C6, C7). Collectively, these benchmarks test an agent's logical consistency and capacity for causal explanation.

**Social Comprehension and Rationale Generation.** This final category of foundational benchmarks evaluates the highest level of cognitive skill: understanding the intentions of other agents and justifying the ego-vehicle's own actions. These capabilities are crucial for ensuring social compliance and building human trust (C7). LingoQA-Bench (Marcu et al., 2024) provides nearly 420,000 action-justification VQA pairs, requiring a model not only to predict an action but also to explain the reason behind it (C4, C7). Similarly, DriveAction-Bench (Hao et al., 2025) employs a tree-structured evaluation framework to assess whether a model's decisions are both human-like and logically sound (C1, C2, C5, C6, C7). These benchmarks collectively push models beyond being passive observers of the environment towards becoming active, explainable, and socially aware participants in traffic.

### 6.1.2 Evaluating System-Level and Interactive Performance

While foundational benchmarks are essential for assessing cognitive skills in isolation, the true viability of a driving agent can only be determined by evaluating its performance "in the loop," where its actions have direct and immediate consequences. This section reviews the platforms and datasets designed for this stage of validation.

**Closed-Loop System Validation.** System-level evaluation ensures that an agent's high-level reasoning translates into physically executable and effective low-level control. This directly tests Decision-Reality Alignment (C4). Benchmarks in this category provide comprehensive long-duration data that enable validation of the entire system stack, from perception to action. An example is DriveMLM (Guo et al., 2024), which offers a large-scale dataset of 280 hours of driving data with multi-modal inputs (C1, C4, C6). LaMPilot-Bench (Ma et al., 2024b) integrates large language models into autonomous driving systems by generating executable code as driving policies, evaluating language-driven autonomous driving agents (C4, C6). The explicit goal of such datasets is to provide the necessary resources to bridge the critical gap between abstract, language-based decisions and the continuous control signals required for safe driving.

**Interactive and Critical Scenarios.** Beyond ensuring internal coherence, robust evaluation must also assess how the agent performs when interacting with other dynamic agents, especially in safety-critical situations. The focus here shifts from general driving to the agent's ability to adapt and react under pressure. Bench2ADVLM-Bench (Zhang et al., 2025a) exemplifies this approach by offering a dedicated benchmark specifically designed for closed-loop testing in 220 curated "threat-critical" scenarios (C1, C2, C4, C5, C6, C7). This scenario-based methodology, often operationalized within interactive simulation platforms, is essential for rigorously assessing a model's performance and adaptability when safety is on the line.

### 6.1.3 Evaluating Robustness and Trustworthiness

Building upon the assessment of system performance in standard interactive settings, this final stage of evaluation moves to the most demanding frontier: assessing an agent's behavior under stress, in the face of novelty, and against its own internal biases. These benchmarks are designed to move beyond average-case performance and probe the limits of a system's reliability, which is a prerequisite for establishing public trust and ensuring safe deployment.

**Evaluating Generalization to the Long Tail.** A fundamental limitation of any data-driven model is its performance on events that are rare in its training distribution. We review benchmarks designed specifically to measure robustness against such long-tail scenarios, directly addressing a core challenge for autonomous driving (C5). These datasets provide a controlled environment for testing a model's ability to generalize its reasoning to novel situations. For example, as shown in Fig. 8 (c), DrivingDojo (Wang et al., 2024e) is an interactive dataset curated with thousands of video clips containing rare events like crossing animals and falling debris, as well as complex multi-agent interactions such as cut-ins (C2, C5, C6, C7). Similarly, CODA-LM-Bench (Chen et al., 2025) provides extensive annotations for corner cases involving abnormal pedestrian behavior and unique traffic signs (C2, C5, C6, C7). Other works, such as NuPrompt (Wu et al., 2025a), address this challenge by introducing language-prompted 3D object tracking, which tests and improves a model's ability to generalize to unseen or rare attribute combinations (C1,C5).

**Verifying System Trustworthiness and Safety.** Beyond handling external novelty, a reliable agent must also possess intrinsic trustworthiness. This category features benchmarks that move beyond conventional accuracy metrics to systematically evaluate system safety, reliability, and robustness against internal failures. This directly corresponds to the core challenges of Perception-Cognition Bias (C2) and Regulatory Compliance (C6). AutoTrust-Bench (Xing et al., 2025) introduces the first comprehensive benchmark for VLM trustworthiness in driving, establishing a systematic protocol that covers robustness to data corruption, fairness, privacy, and safety evaluation (C1, C2, C5, C6, C7). Complementing this, as shown in Fig. 8 (d), DriveAction-Bench (Xie et al., 2025) focuses specifically on model reliability and corruption robustness by testing the full pipeline under a variety of challenging conditions (C1, C2, C5, C6, C7). Furthermore, other datasets contribute to this theme by design. For example, DrivingVQA (Corbière et al., 2025) (C1, C2, C6, C7) and DriveLMM-o1 (Ishaq et al., 2025) (C1, C4, C6, C7) enhance safety by grounding decisions in explicit traffic rules, while NuScenes-SpatialQA-Bench (Tian et al., 2025b) improves robustness by forcing models to reason from ground-truth sensor data rather than relying on potentially fallible intermediate perception modules (C1, C2).

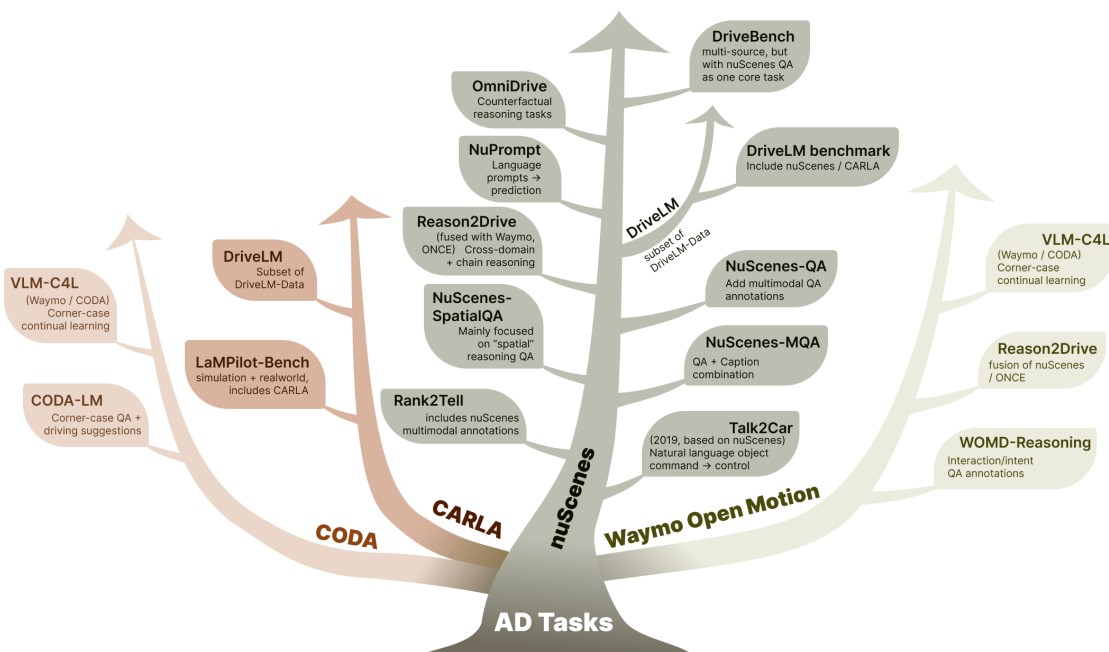

Figure 9: A genealogical chart of the autonomous driving benchmark ecosystem. We illustrate the derivative and inheritance relationships among key datasets, highlighting how foundational platforms (e.g., NuScenes) have inspired a "family" of specialized benchmarks designed to evaluate specific reasoning capabilities.

## 6.2 Comprehensive Benchmark Dataset Landscape

To synthesize the preceding categorical discussion and provide a high-level, comparative view, this subsection presents resources that map the evaluation ecosystem. We have constructed a dataset genealogy in Fig. 9. This chart reveals the derivative and inheritance relationships between different benchmarks, illustrating how foundational datasets like NuScenes have spawned an entire "family" of specialized evaluations. This genealogical view, complemented by a comprehensive comparison table (see Table 2 in Appendix B), serves as a quick reference for researchers to navigate the relationships and core attributes of existing benchmarks.

## 6.3 Discussion and Insights: Prevailing Trends

**Prevailing Trends.** Our review of the evaluation-centric landscape highlights several key trends that shape how the field measures progress:

- **From Physical Outcomes to Cognitive Processes:** A clear shift is underway from metrics based on physical outcomes (e.g., collision rates, trajectory error) to evaluations targeting the cognitive process itself. Benchmarks increasingly focus on assessing the quality of rationale generation, the logical consistency of reasoning chains, and the understanding of causality, directly addressing the need for interpretability (C7).
- **From Static VQA to Dynamic Simulation:** The evaluation paradigm is evolving from static, open-loop, question-answering formats (e.g., NuScenes-QA) to dynamic, closed-loop, interactive simulations. This progression is essential for assessing system-level behavior and the critical Responsiveness-Reasoning Tradeoff (C3).
- **Deliberate Curation of the Long Tail:** There is a growing recognition that performance on "average" scenarios is insufficient. A dominant trend is the deliberate and resource-intensive curation of benchmarks (e.g., DrivingDojo, CODA-LM) specifically designed to test generalization and robustness in rare, safety-critical, and Long-tail Scenarios (C5).
- **Emergence of Holistic Trustworthiness:** Evaluation is moving beyond simple task accuracy. New benchmarks (e.g., AutoTrust-Bench) establish comprehensive protocols to measure holistic

trustworthiness, including robustness to data corruption (C2), fairness, and adherence to safety and Regulatory Compliance (C6).

**Cognitive Hierarchy-Informed Evaluation Principles.** A similar stratification can be observed in benchmark design. Egocentric-level datasets such as NuScenes-QA (Qian et al., 2024), DriveLM-Bench (Sima et al., 2024), and Reason2Drive (Nie et al., 2024) emphasize spatial understanding, causal reasoning, and multi-step decision consistency, targeting situational modeling capabilities. In contrast, OmniDrive (Wang et al., 2025b), DrivingDojo (Wang et al., 2024e), and AutoTrust-Bench (Xing et al., 2025) focus on long-tail scenarios, counterfactual reasoning, safety robustness, and rule compliance, probing generalization and norm adherence in complex social contexts. These developments reflect a gradual shift from purely physical metrics toward layered cognitive capability assessment.

From a benchmark construction perspective, the cognitive hierarchy provides a capability-oriented evaluation principle: separating physical feasibility, interaction logic, and social norm reasoning into distinct layers of measurement. Such stratified evaluation reduces ambiguity in mixed metrics and enables clearer mapping between performance gains and specific cognitive improvements.

In summary, the true viability of these system-centric innovations hinges on the ability to rigorously and reliably measure their performance. The development of such evaluation methods is therefore a critical and parallel research frontier, which will be essential for guiding the field toward truly intelligent and deployable autonomous systems.

# 7 Conclusion and Future Work

This survey argues that the progression toward higher levels of vehicle autonomy is fundamentally a problem of reasoning. We introduce a taxonomy that deconstructs this problem into seven core challenges (C1–C7), spanning from Egocentric Reasoning (C1–C4) to the more complicated Social-Cognitive (C5–C7) domain. Our analysis posits that while egocentric challenges represent profound engineering hurdles, the ultimate bottleneck to achieving human-like operational capabilities lies in mastering the social-cognitive challenges.

Our review reveals a clear, accelerating response to this paradigm shift. In system-centric approaches (Sec. 5), research moves decisively from optimizing isolated modules toward holistic, reasoning-centric architectures. The shift toward interpretable "glass-box" agents underscores this trend. In parallel, evaluation-centric practices (Sec. 6) are evolving. Focus is shifting from physical outcomes to assessing cognitive processes, rationale generation, and robustness in curated long-tail scenarios.

Despite progress in methodology and evaluation, our analysis concludes that a fundamental tension remains unresolved. This tension exists between the powerful, deliberative, but high-latency symbolic reasoning offered by large models, and the millisecond-scale, physically-grounded, and safety-critical demands of real-world vehicle control. Reconciling these two worlds (the abstract and the physical, the deliberative and the reactive) is the central, most pressing objective for future AD research. Based on the critical gaps identified in our analysis, we outline the following primary directions for future research that are essential for bridging the gap from reasoning-capable models to road-ready AD agents.

**Verifiable Neuro-Symbolic Architectures.** The Responsiveness-Reasoning Tradeoff (C3) and the Decision-Reality Alignment (C4) gap represent the immediate architectural barriers. Future work should move beyond simple dual-process concepts (i.e., "fast" and "slow" thinking). A critical frontier is developing verifiable neuro-symbolic architectures. These systems must not only provide an arbitration mechanism to manage latency but also offer formal assurances that abstract symbolic plans (e.g., "yield to the merging vehicle") are reliably, safely grounded in the sub-symbolic, continuous control space of the vehicle.

**Robust Reasoning Under Multi-Modal Uncertainty.** A foundational challenge remains in how reasoning engines handle the noisy, asynchronous, and often contradictory data from heterogeneous signals (C1). While many models assume clean inputs, real-world operation demands robustness to the Perception-Cognition Bias (C2). Future research should develop architectures that can explicitly reason about data

uncertainty, fuse conflicting evidence, and perform compensatory inference to maintain a stable world model, especially during sensor degradation or failure in adverse conditions.

**Dynamic Grounding in External Regulatory Knowledge.** True Regulatory Compliance (C6) on a global scale requires more than pre-trained knowledge. Current models lack a mechanism to adapt to the varied legal frameworks of different jurisdictions. Future work should explore "open-world" systems that can dynamically query, retrieve, and interpret knowledge bases of formal traffic laws. A key challenge will be grounding this symbolic, legal reasoning into the agent's real-time decision-making process.

**Generative and Adversarial Evaluation.** Tackling Long-tail Scenarios (C5) and the methodological limitations of evaluation highlight a critical need for new verification and validation paradigms. Current benchmarks, while valuable, are finite and curated, primarily testing against "known unknowns." Future research should invest in generative and adversarial evaluation frameworks. Such systems would leverage simulation and world models not just to test agents, but to automatically discover novel, systemic failure modes, thereby moving the field closer to addressing the challenge of "unknown unknowns."

**Scalable Models for Implicit Social Negotiation.** Mastering the Social Game (C7) remains a major research frontier. Current approaches in both methods and evaluation are largely nascent and often limited to explicit Regulatory Compliance (C6). The next generation of research must move beyond legal compliance to model the fluid, implicit, and non-verbal social game of human interaction. This requires developing scalable models that can infer latent human intent, negotiate multi-agent interactions, and generate behavior that is not only safe but also socially legible and predictable to other road users.

## Broader Impact and Societal Considerations

AD systems may have far-reaching societal impacts beyond technical performance. While this work focuses on a diagnostic and methodological analysis of reasoning capabilities in AD, we briefly situate these technical considerations within their broader societal context.

A primary concern relates to safety and accountability. In real-world deployments, failures of autonomous systems raise difficult questions regarding responsibility and liability, particularly when decision-making processes are opaque or hard to audit. Insufficient reasoning capabilities, especially in rare, ambiguous, or interactive scenarios, can exacerbate such concerns by making system behavior less predictable and harder to justify post hoc. From this perspective, strengthening reasoning is not only a technical objective, but also important for improving transparency, traceability, and auditability (e.g., via clearer decision rationales and system-level logging) in responsible deployment.

Environmental and systemic impacts also warrant consideration. Advances in autonomous driving may lower the barrier to private vehicle usage, potentially encouraging increased car ownership and traffic volume. Even with the adoption of electric vehicles, large-scale manufacturing and energy consumption carry non-negligible environmental costs. Optimizing autonomous driving systems without considering these broader effects may unintentionally reinforce car-centric transportation paradigms rather than more holistic mobility solutions.

Although this work does not directly address policy, ethics, or environmental modeling, it highlights a foundational technical limitation relevant to many of these societal concerns. Specifically, we argue that reasoning has become an increasingly critical bottleneck within the *tightly coupled stack* of perception, prediction, planning, and decision-making in complex and long-tail driving scenarios. Addressing this limitation is important for enabling safer, more reliable, and more responsibly deployable autonomous systems. We hope this perspective encourages future research that jointly considers technical robustness, evaluation practice, and broader societal implications.

## Acknowledgments

This work was partially supported by the Tsinghua SIGS KA Cooperation Fund.

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

# Appendix

# A    Reasoning Paradigms in LLMs and MLLMs

## A.1    Reasoning Paradigms in LLMs

To address complicated challenges in natural language processing, such as answering long-tail questions and performing task planning (Ichter et al., 2022; Yao et al., 2023b), reasoning mechanisms in LLMs decompose problems into a series of manageable, intermediate steps. This is fundamentally achieved by prompting the model to generate an explicit thought process before arriving at a conclusion (Kojima et al., 2022), a technique pioneered by methods like Chain-of-Thought (CoT) (Wei et al., 2022). Building upon this principle, existing reasoning paradigms can be categorized based on the architecture of the generated thought process (Zhang

et al., 2025b). This classification reflects an evolution in methodology designed to handle increasing tasks. These paradigms are broadly grouped into three categories: sequential reasoning, which constructs a single linear path (Zhou et al., 2023; Yao et al., 2023b); structured reasoning, which explores multiple paths in parallel (Wang et al., 2023; Yao et al., 2023a); and iterative reasoning, which refines a solution through feedback loops (Madaan et al., 2023; Shinn et al., 2023b).

**Sequential Reasoning.** Sequential reasoning represents the foundational paradigm, employing a linear, unidirectional process to generate a single, continuous chain of thought (Yao et al., 2023b; Creswell et al., 2023). This approach emulates human-like step-by-step problem-solving and serves as the basis for more advanced reasoning structures (Nye et al., 2021). The most direct implementation is the CoT method (Wei et al., 2022), which instructs the model to produce a linear thought process, often activated through simple prompts (e.g., "Let's think step by step") (Kojima et al., 2022) or few-shot exemplars. Subsequent research has focused on enhancing the robustness and reliability of this linear path. Strategies include ensemble approaches such as Self-Consistency (Wang et al., 2023) and CoRe (Zhu et al., 2023), which select the most consistent answer from multiple reasoning chains, and self-correction methods such as Chain of Verification (CoV) (Dhuliawala et al., 2024), which introduce an explicit step to verify and refine the generated reasoning.

**Structured Reasoning.** To address complicated problems that require exploration beyond a single line of thought, structured reasoning transcends the limitations of linear approaches (Wei et al., 2022; Kojima et al., 2022) by concurrently exploring multiple reasoning paths (Wang et al., 2023; Creswell et al., 2023). This paradigm organizes these paths within non-linear data structures, such as trees or graphs, enabling a model to deliberately consider, evaluate, and integrate diverse lines of thought (Shinn et al., 2023a; Hao et al., 2023). An extension of the linear chain is the tree structure, implemented in the Tree of Thoughts (ToT) framework (Yao et al., 2023a), which expands a single thought chain into a tree of intermediate states (Ding et al., 2025b; Wu et al., 2025b). A foresight module then evaluates each branch to systematically navigate the solution space. To facilitate more sophisticated reasoning, graph-based structures like Graph of Thoughts (GoT) (Besta et al., 2024) advance beyond trees by allowing paths to not only branch but also merge (Jin et al., 2024). The capacity to synthesize information from different branches provides graph-based reasoning with superior flexibility for problems that demand a holistic analysis.

**Iterative Reasoning.** In contrast to the generative and exploratory nature of the preceding paradigms, iterative reasoning introduces a feedback loop for refinement and self-correction (Shinn et al., 2023a; He et al., 2025; Yu et al., 2025). This approach targets higher solution accuracy and robustness (Yao et al., 2023a), operating on a generate-and-refine cycle where an initial solution undergoes progressive improvement. This paradigm can be categorized by the source of feedback. The fundamental form uses an internal loop, where a model critiques and revises its own output, as seen in Self-Refine (Madaan et al., 2023). A more advanced form incorporates external feedback from tools like search engines, as exemplified by the ReAct framework (Yao et al., 2023b), which interleaves thought, action, and observation. A third form leverages experiential feedback for long-term improvement. Methods like Chain of Hindsight (CoH) (Liu et al., 2024a) analyze past performance on a task to generate an improved reasoning path, which is then used to guide the model on similar future tasks, establishing a meta-level learning loop (Shinn et al., 2023a).

## A.2 Reasoning Paradigms in MLLMs

The paradigm of reasoning enables LLMs to perform complicated logical inference (Wei et al., 2022). For this intelligence to become applicable in the physical world, however, reasoning must transcend purely linguistic symbols and integrate with sensory modalities such as vision (Zitkovich et al., 2023). MLLMs are pivotal in achieving this integration. Applying reasoning within MLLMs extends the capabilities of these models beyond abstract textual processing, allowing them to perceive and understand the physical world directly (Yang et al., 2023). By transforming raw visual pixels into semantic concepts, MLLMs facilitate complex reasoning grounded in visual information (Zhang et al., 2024d; Pham & Ngo, 2025).

In the course of the development of multimodal reasoning, numerous models have emerged, establishing new paradigms for applications. For example, in reasoning-based segmentation, models no longer depend on predefined semantic labels. Instead, these models can interpret and execute complex instructions, such as

"segment the fruit in the image with the highest vitamin C content," a task that requires both functional understanding and world knowledge (Lai et al., 2024; Xia et al., 2024; Liu et al., 2025d; 2024b; Wei et al., 2025; 2024a). Furthermore, reasoning capabilities have expanded from two-dimensional visual understanding to three-dimensional scene comprehension (Hong et al., 2023). In 3D multimodal reasoning, models integrate data from multi-view images, depth maps, and point clouds (Guo et al., 2023). These inputs enable the models to infer spatial structures, manage occlusions, and reason about physical feasibility (Azuma et al., 2022). Through the incorporation of explicit spatial representations and 3D encoders into language models, these systems can handle complex physical reasoning tasks, such as "determining whether a robotic arm can grasp an object without colliding with others" (Zhao et al., 2025; Huang et al., 2025; Driess et al., 2023; Zitkovich et al., 2023).

Currently, cutting-edge multimodal models, including the Qwen-VL series (Bai et al., 2025), Google Gemini (Anil et al., 2023), and GPT-5 (OpenAI, 2023), significantly advance these capabilities. The reasoning mechanisms and cross-modal spatial modeling inherent to these models are instrumental in bridging the gap between 2D perception and 3D understanding (Hong et al., 2023). This progress lays the foundation for intelligent agents capable of interacting with the physical world and comprehending complex scenes (Zitkovich et al., 2023). Such spatial reasoning abilities also provide crucial technical support for domains like AD, where the real-time understanding of dynamic 3D environments is essential (Xu et al., 2024b).

## B   Methods and Benchmark Taxonomy

The two tables below correspond to Sec. 5 and Sec. 6, respectively, providing a quick reference for researchers.

### B.1   Methods Taxonomy

Table 1 summarizes all methods in Sec. 5 in chronological order, along with their classification in Sec. 5, the corresponding challenges in Fig. 4, and the characteristics of each method.

Table 1: A comprehensive overview and comparison of key methods for autonomous driving reasoning. This table summarizes the primary attributes of each method, including its publication **Date**, **Affiliation**, core reasoning **Challenges** (C1–C7) it addresses, associated **Keywords**.

| Date | Methods | Affiliation | Challenges | Keywords |
|---|---|---|---|---|
| 2023.07 | Drive like a human (Fu et al., 2024) | Reasoning in Planning and Decision-Making | C5, C7 | Human-like driving; Social reasoning |
| 2023.10 | Driving with LLMs (Chen et al., 2024) | Holistic Architectures: Integrated and End-to-End Agents | C1, C4 | Object–language fusion; Explainable planning |
| 2023.12 | Dolphins (Ma et al., 2024a) | Reasoning in Planning and Decision-Making | C1, C4, C5 | Multimodal CoT; Adaptive behavior |
| 2023.12 | GPT-Driver (Mao et al., 2023) | Holistic Architectures: Integrated and End-to-End Agents | C4, C5 | Language justification; Glass-box driving |
| 2024.01 | Think-Driver | Reasoning in Planning and Decision-Making | C1, C4, C5 | Explicit CoT; Interpretable planning |

*Continued on next page*

| Date | Methods | Affiliation | Challenges | Keywords |
|---|---|---|---|---|
| | (Zhang et al., 2024b) | | | |
| 2024.02 | DiLu (Wen et al., 2024) | Reasoning in Planning and Decision-Making | C1, C4, C5 | Memory reasoning; Self-reflection |
| 2024.03 | DriveCoT (Wang et al., 2024b) | Holistic Architectures: Integrated and End-to-End Agents | C4, C5 | End-to-end CoT; Reasoning supervision |
| 2024.06 | Optimizing AD for Safety (Sun et al., 2024) | Reasoning in Planning and Decision-Making | C6, C7 | RLHF safety; Human feedback |
| 2024.06 | DriveVLM (Tian et al., 2024c) | Reasoning in Planning and Decision-Making | C3, C4 | Hybrid VLM; Reasoning planning |
| 2024.06 | Editable Scene Simulation (ChatSim) (Wei et al., 2024c) | Reasoning in Supporting and Auxiliary Tasks | C3, C5, C7 | Language editing; Controllable simulation |
| 2024.07 | TOKEN (Tian et al., 2024a) | Reasoning in Planning and Decision-Making | C1, C4, C5 | Object tokens; Structured reasoning |
| 2024.07 | Reason2Drive (Nie et al., 2024) | Reasoning-Enhanced Perception | C1, C5 | Reasoning dataset; Perception benchmark |
| 2024.08 | Hybrid Reasoning (Azarafza et al., 2024) | Holistic Architectures: Integrated and End-to-End Agents | C1, C2 | Symbolic–neural fusion; Robust reasoning |
| 2024.09 | Crash Severity Analysis (Zhen et al., 2024) | Reasoning in Supporting and Auxiliary Tasks | C4, C5 | Accident reasoning; Causal analysis |
| 2024.11 | DriveGPT4 (Xu et al., 2024b) | Holistic Architectures: Integrated and End-to-End Agents | C4, C5 | Multimodal GPT; End-to-end reasoning |
| 2024.12 | PKRD-CoT (Luo et al., 2024) | Reasoning in Planning and Decision-Making | C1, C4, C5 | Knowledge CoT; Robust planning |
| 2024.12 | VLM-RL (Huang et al., 2024) | Reasoning in Planning and Decision-Making | C3, C4 | VLM rewards; RL alignment |

| Date | Methods | Affiliation | Challenges | Keywords |
|---|---|---|---|---|
| 2025.01 | DriveLM (Sima et al., 2024) | Holistic Architectures: Integrated and End-to-End Agents | C1, C4, C5 | Graph reasoning; Integrated planning |
| 2025.02 | TeLL-Drive (Xu et al., 2025a) | Reasoning in Supporting and Auxiliary Tasks | C4, C5 | Teacher–student; Reasoning distillation |
| 2025.03 | TRACE (Puthumanaillam et al., 2025) | Reasoning-Informed Prediction | C2, C4, C5 | Tree-of-Thought; Counterfactual prediction |
| 2025.03 | CoT-VLM4Tar (Ren et al., 2025) | Holistic Architectures: Integrated and End-to-End Agents | C1, C4, C5 | Abnormal traffic; CoT reasoning |
| 2025.03 | Driving with Regulation (Cai et al., 2024) | Reasoning in Planning and Decision-Making | C4, C6 | Regulation grounding; Compliant planning |
| 2025.03 | HiLM-D (Ding et al., 2025a) | Reasoning-Enhanced Perception | C1, C2 | Two-stream perception; Risk object reasoning |
| 2025.03 | WOMD-Reasoning (Li et al., 2024c) | Reasoning-Informed Prediction | C5, C7 | Rule grounding; Knowledge supervision |
| 2025.03 | ORION (Fu et al., 2025) | Holistic Architectures: Integrated and End-to-End Agents | C3, C4 | Knowledge supervision; Generative planning |
| 2025.03 | CoT-VLA (Zhao et al., 2025) | Holistic Architectures: Integrated and End-to-End Agents | C3, C4 | VLA CoT; Reason–action alignment |
| 2025.03 | ARGUS (Man et al., 2025) | Reasoning-Enhanced Perception | C1, C2 | Box-guided reasoning; Visual grounding |
| 2025.04 | V2V-LLM (Chiu et al., 2025) | Holistic Architectures: Integrated and End-to-End Agents | C1, C5, C7 | Vehicle communication; Cooperative reasoning |
| 2025.04 | OmniDrive | Holistic Architectures: Integrated and End-to-End Agents | C1, C3, C5 | Counterfactual evaluation; Long-tail reasoning |

*Continued on next page*

| Date | Methods | Affiliation | Challenges | Keywords |
|------|---------|-------------|------------|----------|
| | (Wang et al., 2024a) | | | |
| 2025.06 | X-Driver (Liu et al., 2025c) | Holistic Architectures: Integrated and End-to-End Agents | C1, C4, C5 | Autoregressive decisions; Closed-loop control |
| 2025.06 | HMVLM (Wang et al., 2025a) | Reasoning in Planning and Decision-Making | C3, C4 | Hierarchical CoT; Multi-stage reasoning |
| 2025.06 | Drive-R1 (Li et al., 2025a) | Reasoning in Planning and Decision-Making | C3, C4 | CoT-guided RL; Policy refinement |
| 2025.07 | Controllable Traffic Simulation (Liu et al., 2024c) | Reasoning in Supporting and Auxiliary Tasks | C3, C5, C7 | Scenario generation; Language control |
| 2025.08 | VLM-AD (Xu et al., 2024a) | Reasoning in Planning and Decision-Making | C1, C4 | VLM teacher; Reasoning labels |
| 2025.09 | OCCVLA (Liu et al., 2025b) | Reasoning-Enhanced Perception | C1, C3 | Occupancy reasoning; 3D semantics |
| 2025.09 | EMMA (Hwang et al., 2025) | Holistic Architectures: Integrated and End-to-End Agents | C1, C3, C5 | Multimodal fusion; Multimodal fusion |
| 2025.09 | AgentThink (Qian et al., 2025) | Reasoning in Planning and Decision-Making | C1, C4, C5 | Tool calling; Open-world reasoning |
| 2025.10 | Poutine (Rowe et al., 2025) | Reasoning in Planning and Decision-Making | C2, C5 | Preference learning; Safety alignment |
| 2025.11 | DMAD (Shen et al., 2025) | Reasoning-Enhanced Perception | C1 | MotionSemantic Decoupling; Structured Perception Learning |
| 2026.01 | VLAAD-TW (Kondapally et al., 2026) | Reasoning in Supporting and Auxiliary Tasks | C1, C5 | Spatiotemporal Reasoning; Selective Attention Fusion |

Table 2: A comprehensive overview and comparison of key benchmarks for autonomous driving reasoning. This table summarizes the primary attributes of each benchmark, including its publication **Date**, core reasoning **Challenges** (C1–C7) it addresses, associated **Keywords**, **Data Scale**, and community engagement metrics (GitHub **Stars**). Repository **Links** are also provided for reference.

| Date | Dataset | Challenges | Keywords | Data Scale | Stars | Link |
|---|---|---|---|---|---|---|
| 2023.08 | RoadTextVQA (Tom et al., 2023) | C1,C2,C5,C6 | Driving VideoQA; Scene Text; Road Signs | 3,222 videos / 10,500 QA | 5 | ⬤ |
| 2023.12 | DriveMLM (Guo et al., 2024) | C1,C4,C6 | Closed-loop Driving; Multi-modal Input; Decision Alignment | 280h driving / 50k routes / 30 scenarios / 4 cams + LiDAR | 179 | ⬤ |
| 2024.01 | Rank2Tell (Sachdeva et al., 2024) | C1,C4,C6,C7 | Importance Ranking; Language Explanations; Ego-centric | 116 clips / 3 cams + LiDAR + CAN | - | - |
| 2024.01 | NuScenes-MQA (Inoue et al., 2024) | C1 | 3D VQA; Markup-QA; Spatial Reasoning | 1.46M QA / 34k scenarios / 6 cams | 31 | ⬤ |
| 2024.02 | NuScenes-QA-Bench (Qian et al., 2024) | C1,C5 | Multi-modal VQA; 3D Scene Graph | 34K scenes / 460K QA | 210 | ⬤ |
| 2024.06 | LaMPilot-Bench (Ma et al., 2024b) | C4,C6 | Language Model Programs ; Driving Instructions; Code-as-Policy | 4,900 scenes (500 test) | 31 | ⬤ |
| 2024.09 | DriveLM-Bench (Sima et al., 2024) | C1,C4,C5,C7 | Graph VQA; End-to-end Autonomous Driving | 19,200 frames / 20,498 QA pairs | 1.2k | ⬤ 🤗 |
| 2024.09 | LingoQA-Bench (Marcu et al., 2024) | C4,C7 | ActionJustification VQA; End-to-End Benchmark | 28K scenarios / 419.9K QA pairs | 183 | ⬤ |
| 2024.09 | Reason2Drive-Bench (Nie et al., 2024) | C1,C4,C5 | Chain-based Reasoning; Cross Datasets | 600K+ video-text pairs | 92 | ⬤ |
| 2024.12 | DrivingDojo (Wang et al., 2024e) | C2,C5,C6,C7 | Driving World Model; Interactive Dataset | 18k videos | 73 | ⬤ 🤗 |
| 2024.12 | AutoTrust-Bench (Xing et al., 2025) | C1,C2,C5,C6,C7 | Trust/safety/robustness/privacy/fairness | 10K+ driving scenes; 18K+ QA pairs | 48 | ⬤ |
| 2025.01 | DrivingVQA (Corbière et al., 2025) | C1,C2,C6,C7 | VQA; Chain-of-Thought | 3,931 multiple-choice problems | 0 | ⬤ 🤗 |
| 2025.02 | CODA-LM-Bench (Chen et al., 2025) | C2,C5,C6,C7 | Corner cases; LVLM evaluation | 9,768 scenarios; over 63K annotations | 92 | ⬤ 🤗 |
| 2025.02 | NuPrompt (Wu et al., 2025a) | C1,C5 | Language Prompt; 3D Perception; Multi-Object Tracking (MOT) | 40.1K language prompts / avg. 7.4 tracklets per prompt | 149 | ⬤ |
| 2025.03 | DriveLMM-o1 (Ishaq et al., 2025) | C1,C4,C6,C7 | Step-by-Step Reasoning | 18k train / 4k test QA | 66 | ⬤ 🤗 |
| 2025.04 | NuScenes-SpatialQA-Bench (Tian et al., 2025b) | C1,C2 | Spatial Reasoning | 3.5M QA | 18 | ⬤ 🤗 |
| 2025.06 | STSBench-Bench (Fruhwirth-Reisinger et al., 2025) | C1,C5,C7 | Spatio-temporal Reasoning | 971 MCQs / 43 scenes | 10 | ⬤ 🤗 |
| 2025.06 | DriveAction-Bench (Hao et al., 2025) | C1,C2,C5,C6,C7 | Tree-structured Evaluation; Human-like Decisions | 16,185 QA / 2,610 scenes | - | 🤗 |
| 2025.06 | OmniDrive (Wang et al., 2025b) | C1,C4,C5,C6 | Counterfactual reasoning | - | 486 | ⬤ |
| 2025.07 | WOMD-Reasoning (Li et al., 2024b) | C1,C4,C5,C6,C7 | Interaction Reasoning; Traffic Rules | 3M QA | 37 | ⬤ |
| 2025.08 | Bench2ADVLM-Bench (Zhang et al., 2025a) | C1,C2,C4,C5,C6,C7 | Closed-loop evaluation; Dual-system adaptation | 220 threat-critical scenarios | 2 | ⬤ |
| 2025.10 | DriveBench-Bench (Xie et al., 2025) | C1,C2,C5,C6 | Reliability; Corruption Robustness | 19,200 frames / 20,498 QA | 112 | ⬤ 🤗 |
| 2025.12 | SURDS-Bench (Guo et al., 2025) | C1,C2 | Spatial Reasoning; Spatial Understanding; Driving VQA | 41,080 train QA/ 9,250 test QA | 54 | ⬤ |

## B.2 Benchmark Taxonomy

Table 2 summarizes all benchmarks in Sec. 6 in chronological order, including their dataset characteristics and the corresponding challenges in Sec. 4. In addition, some GitHub and Hugging Face links are provided at the end for quick reference and navigation of the readers.

