# OpenReview forum: "A‌ Survey of Reasoning in Autonomous Driving Systems: Open Challenges and Emerging Paradigms"
_TMLR — Accepted by TMLR_

### Review · Reviewer_Xqg8 · 2025-12-24

**Summary Of Contributions:**

The paper argues that reasoning is the primary bottleneck for the large-scale deployment of autonomous vehicles (AVs). To support this position, the authors propose a three-level cognitive hierarchy that decomposes driving tasks by increasing reasoning complexity: (1) Sensorimotor Level, (2) Egocentric Reasoning Level, and (3) Social-Cognitive Level. Based on this hierarchy, the paper introduces a taxonomy of seven challenges in autonomous driving that require reasoning. The authors then survey existing methods that incorporate reasoning, particularly those leveraging LLMs and MLLMs, within AV systems, and review benchmarks designed to evaluate reasoning capabilities. The paper concludes with a discussion of open challenges and future research directions.

Key strengths:
- Section 5 provides a thorough overview of current approaches to reasoning in AVs, and the summary of the prevailing trends was interesting.
- The taxonomy of reasoning challenges in AVs is reasonable.
- Table 1 is a useful and comprehensive overview of benchmarks, and I appreciated that the authors tied them back to the 7 proposed challenges.

Key weaknesses:
- The paper is overly long, unfocused, and conceptually confusing. The proposed cognitive hierarchy is confusing and internally inconsistent (see points below for more details). Existing surveys on related topics, such as MLLMs for autonomous driving (A Survey on Multimodal Large Language Models for Autonomous Driving, Cui et al. WACV Workshops, 2025) and Chain-of-thought for autonomous driving (Chain-of-Thought for Autonomous Driving: A Comprehensive Survey and Future Prospects, Cui et al. ArXiv 2025) are more concise, better scoped, and easier to digest.
- Most of the figures (e.g., Figures 6, 7, 8, 9, 11, 13, 14) appear to be copied verbatim from prior papers, which is not acceptable.
- Many references are inappropriate or misleading given the context (see points below for more details).

**Additional Comments:**

None.

**Audience:**

Yes

**Audience Explanation:**

Reasoning is playing an increasingly important role in autonomous driving systems, and AV researchers will find certain parts of this paper useful and interesting, especially the taxonomy of reasoning challenges and the section on benchmarks for evaluating reasoning capabilities.

**Broader Impact Concerns:**

None.

**Claims And Evidence:**

No

**Claims Explanation:**

The central claim that reasoning is the primary bottleneck in autonomous driving is not convincingly justified, and several issues throughout the paper weaken the argument.

While it is reasonable to argue that reasoning plays an increasingly important role, the paper does not provide sufficient evidence to support the stronger claim that perception and control are no longer dominant bottlenecks. In practice, failures in autonomous driving systems often arise from tightly coupled limitations across perception, prediction, planning, and reasoning, rather than from reasoning alone.

Additional concerns:
- The distinction between perception and reasoning is often unclear or misleading. Reasoning is crucial within perception itself (e.g., for handling rare or long-tail objects), yet the paper treats them as largely separable components.
- The cognitive hierarchy is confusing and internally inconsistent. Many tasks assigned to the “Social-Cognitive Level” (e.g., reasoning about agent intentions) could reasonably be classified as egocentric reasoning. The terminology (e.g., “vehicle-to-environment,” “vehicle-to-agents,” “vehicle-in-society”) further confuses rather than clarifies these distinctions.
- Figure 1 does not clearly motivate the need for reasoning.  It’s unclear what a “Current AD System” is and why it fails on these scenarios. For example, in E2 and E3, it seems to be reasoning (“The vehicle can turn left since it’s on the major road.”, “The gap is sufficient, so the vehicle can merge to the right.”). It's therefore unclear how to distinguish the "Current AD System" from the "Reasoning in AD" system.
- Several citations are inappropriate or misleading, including references that are unrelated to reasoning in autonomous driving or are used in a confusing context (see requested changes). This is particularly problematic given that this is a survey paper.
- Most of the figures (e.g., Figures 6, 7, 8, 9, 11, 13, 14) appear to be copied verbatim from prior papers, which is not acceptable.

**Requested Changes:**

- Clarify and refine the central claim, avoiding the unsupported assertion that reasoning is the primary bottleneck.
- The paper should have more original figures. The majority of figures in a survey paper should not be copied verbatim from other papers.
- Substantially shorten the paper. For example, the background sections on generic LLM/MLLM reasoning paradigms can be significantly shortened or even removed. The open questions in Section 5.3 largely overlap with the reasoning challenges (eg, Reconciling Deliberation and Reaction Time is equivalent to the Responsiveness-Reasoning Tradeoff, the Symbolic-to-Physical Grounding Gap is equivalent to the Decision-Reality Alignment, etc) and are therefore redundant. Survey of current work on reasoning-enabled AV systems does not appear until page 14, reflecting that this survey is unnecessarily long.
- Clarify the distinction between reasoning and perception: The paper treats perception and reasoning as largely distinct, despite reasoning being essential for perception tasks such as handling long-tail or ambiguous visual inputs.
- Rework or simplify the cognitive hierarchy, ensuring that categories are consistent and intuitively defined. For example, I would argue that some of the tasks in the “Social-Cognitive Level” (Level 3) are forms of “Egocentric reasoning” (Level 2), such as reasoning about agent intentions. Moreover, the terms “Vehicle-to-environment”, “vehicle-to-agents”, and “vehicle-in-society” only add to the confusion. Shouldn’t “Reasoning about Agent Intentions” (Level 3 task) by definition be a “Vehicle-to-agents” task (Level 2)?
- Revise or remove confusing figures, particularly Figure 1. The wording of Example E2 is quite confusing: “There is a vehicle at the intersection wants to go straight, while the vehicle wants to go straight as well.” And why are the challenges out of order in Figure 1? More importantly, clarify the difference between a "Current AD System" and a "Reasoning in AD" system.
- Improve overall focus, prioritizing reasoning in autonomous driving rather than providing a broad and generic overview of LLM reasoning.
- Correct citation issues, removing irrelevant references and clarifying misleading ones. For example, the first citation (Oviedo-Trespalacios et al., 2019) has nothing to do with autonomous driving. The citation following the statement about the importance of "robust and generalizable reasoning" (He & Lv, 2023, page 1) has nothing to do with reasoning. (Bojarski et al. 2016) is cited following a sentence about modular pipelines, despite being one of the seminal works on E2E driving. (Rowe et al. 2025) is incorrectly classified as a reasoning for prediction method when it is a reasoning for planning method. The authors need to evaluate their choice of citations more carefully, especially as this is a survey paper.

---

> ### Author Response · Authors · 2026-02-14
> **Response to Reviewer Xqg8 (Part 1/2)**
>
> Thank you very much for your thorough and candid review. Your comments have helped us re-examine our central claims, conceptual framing, figures, and citation practices in a much more rigorous manner. Below, we respond to each major concern in turn.
>
> **Q1:**
> The central claim that reasoning is the primary bottleneck in autonomous driving is not sufficiently justified, and perception/control may remain dominant constraints.
>
> **A1:**
> Thank you for raising this critical concern regarding our central claim. We fully agree that, in real-world autonomous driving systems, failures often arise from coupled limitations across perception, prediction, planning, and reasoning modules, rather than from a single isolated factor. Accordingly, we have revised our core statement from an "exclusive primary bottleneck" framing to a more balanced position: reasoning is one of the key bottlenecks within a coupled system perspective, thereby avoiding conflict with practical system engineering realities. We also explicitly clarify that other modules are not "fully solved"; rather, we emphasize that reasoning capabilities are becoming increasingly critical in complex, long-tail, and socially interactive scenarios.
>
> The specific revisions include:
>
> 1. In Section 1, we replaced "primary bottleneck" with "a growing bottleneck among coupled factors," and clarified that by "reasoning" we refer to explicit semantic reasoning, rule/common-sense utilization, intention modeling in multi-agent interactions, and interpretable decision-making (rather than artificially separating reasoning from perception).
> 2. In Section 3.4, we strengthened the literature-based argument by incorporating discussions from existing surveys and representative works, supporting the claim that as foundational perception capabilities mature, structured reasoning in complex scenarios is emerging as a more prominent constraint.
> 3. In Section 2, we added a boundary clarification stating that this paper does not claim perception/control problems are solved. Instead, we argue that in complex and long-tail scenarios, insufficient reasoning capacity can amplify perception errors, rule conflicts, and interaction uncertainty, thereby becoming one of the key limiting factors in current AD system deployment.
>
> We sincerely appreciate your comments, which prompted us to articulate this claim in a more rigorous and measured manner.
>
> **Q2:**
> The paper is overly long and unfocused; background sections are generic and Section 5.3 overlaps with earlier challenges.
>
> **A2:**
> Thank you for this critical feedback. We have systematically streamlined and structurally refined the manuscript: removing or merging repetitive expressions; moving background material on general LLM/MLLM reasoning paradigms to the appendix to reduce main-text length and enhance thematic focus; and deleting or consolidating content in Section 5.3 that substantially overlapped with the earlier seven challenges.
> After revision, the main text is more tightly centered on the core theme of "reasoning in autonomous driving," while preserving structural completeness. We appreciate you highlighting this important issue.
>
> **Q3:**
> Many figures appear to be copied verbatim from prior work, which is inappropriate for a survey paper.
>
> **A3:**
> Thank you for raising this. We agree that a survey should prioritize original, integrative visualizations. In the revision, we redrew and reorganized the previously heavily referenced figures (notably in Sections 5–6). In particular, Figure 6 and Figure 8 are now reconstructed around our taxonomy and logical organization, rather than reused from prior publications. This addresses the duplication concern and improves coherence between the figures and our framework.
>
> **Q4:**
> Several citations are inappropriate, misleading, or weakly connected to the surrounding claims.
>
> **A4:**
> Thank you for your careful review of our citation practices.
>
> We acknowledge that some references provided only indirect or weak support, which may have led to contextual ambiguity.
>
> For example:
>
> - Regarding the citation of Bojarski et al. (2016), our original intention was to indirectly illustrate the contrast and limitations of modular structures through end-to-end approaches. However, we agree that this citation was not sufficiently precise.
> - Regarding the citation of Oviedo et al. (2019), we intended to use macro-level driving background to introduce the development context of autonomous driving. However, we agree that the reference is not directly related to AD.
>
> Accordingly, in the revised manuscript, we have:
>
> - Removed citations not directly related to the reasoning theme;
> - Re-checked that references supporting key claims are strictly aligned with the surrounding arguments;
> - Corrected isolated misclassification issues (e.g., incorrect methodological categorization).
>
> As a survey paper, we fully agree that citation rigor is particularly important. We appreciate you drawing attention to this issue.

---

> > ### Author Response · Authors · 2026-02-14
> > **Response to Reviewer Xqg8 (Part 2/2)**
> >
> > **Q5:**
> > The distinction between perception and reasoning is unclear; reasoning is also essential within perception itself.
> >
> > **A5:**
> > Thank you for raising this important conceptual concern.
> >
> > We clarify that the distinction between "perception" and "reasoning" in this paper is primarily analytical rather than a strict system-boundary separation.
> >
> > - In end-to-end models, perception and reasoning are typically implicitly coupled;
> > - In modular systems, they are relatively decoupled at the architectural level.
> >
> > As discussed in Section 4.1 and Section 4.2:
> >
> > "The core challenge lies in fusing this diverse information to support complex high-level reasoning."
> >
> > "Overcoming these challenges requires a robust reasoning mechanism capable of cross-modal validation, compensatory inference, and reality-checking to maintain a stable and accurate understanding of the environment."
> >
> > Reasoning can function as a validation and compensation mechanism for perceptual reliability, reflecting a relationship of coupling rather than separation.
> >
> > We further clarify that our emphasis is on high-level structured reasoning in complex decision-making and social interaction, rather than denying the importance of reasoning processes within perception itself.
> >
> > We appreciate your comment, which helped us refine these conceptual boundaries.
> >
> > **Q6:**
> > The cognitive hierarchy is confusing and internally inconsistent; distinctions between Level 2 and Level 3 are unclear.
> >
> > **A6:**
> > Thank you for this in-depth critique of the cognitive hierarchy.
> >
> > We clarify that:
> >
> > - The cognitive hierarchy is an analytical framework intended to characterize increasing cognitive complexity and abstraction;
> > - It is not designed as a mutually exclusive task taxonomy;
> > - A complex scenario may involve capabilities across multiple cognitive levels.
> >
> > For instance, scenarios involving multi-agent intention inference (primarily associated with Level 2) may include egocentric decision reasoning as well as social-norm reasoning components.
> >
> > To clarify the distinction between Level 2 and Level 3, we revised the core criteria from a vague notion of "appearing more advanced" or "involving others" to two more operational axes:
> >
> > (i) Whether the scenario involves reciprocal inference and strategic negotiation in multi-agent interaction;
> >
> > (ii) Whether it requires explicit modeling of social norms, traffic regulations, or implicit negotiation conventions.
> >
> > For boundary cases such as intention reasoning, we added a footnote clarifying that it may function either as an egocentric predictive enhancement or, in more complex negotiation contexts, as a Social-Cognitive capability—thus avoiding rigid categorization.
> >
> > The hierarchy is intended to help researchers understand differences in capability dimensions, rather than to enforce a unique task classification.
> >
> > We appreciate your comments, which prompted us to clarify the positioning and boundaries of this framework.
> >
> > **Q7:**
> > Figure 1 and other figures are confusing; the distinction between "Current AD System" and "Reasoning in AD" is unclear.
> >
> > **A7:**
> > Thank you for pointing out the lack of clarity in Figure 1 and related visualizations.
> >
> > In the revised manuscript, we have:
> >
> > - Rewritten portions of the example descriptions to eliminate linguistic ambiguity;
> > - Adjusted the figure structure to ensure more consistent logical ordering;
> > - Explicitly clarified that "Current AD System" refers to conventional paradigms primarily relying on rule-based or shallow modular decision structures;
> > - Strengthened the explanation of "Reasoning in AD" as emphasizing explicit structured reasoning capabilities.
> >
> > We also revised Example E2, replacing it with a power-outage scenario to illustrate how some AD systems encounter rule-based bottlenecks in such situations, whereas reasoning mechanisms could enable successful navigation.
> >
> > We appreciate your feedback, which has helped improve the clarity of our presentation.
> >
> > Once again, we sincerely thank you for your rigorous and professional critique. Your feedback has prompted systematic improvements in our central claims, structural organization, conceptual boundaries, figure design, and citation rigor. We believe the revised manuscript demonstrates substantially improved logical clarity and academic rigor. All related modifications are highlighted in blue in the revised version.

---

> ### Author Response · Authors · 2026-02-25
> **Kind Follow-Up to Reviewer Xqg8**
>
> Dear Reviewer Xqg8,
>
> Thank you again for your careful review and for the time you have already devoted to our submission. We understand this is a busy period, and we just wanted to kindly follow up in case you have had a chance to look over our response and the revised manuscript. If you have any additional comments or suggestions, we would greatly appreciate your feedback, as it would help us further improve the paper.
>
> Thank you again for your time and consideration.

---

> ### Comment · Reviewer_Xqg8 · 2026-02-26
> **Good rebuttal**
>
> Thank you for your thoughtful rebuttal and for addressing each of my concerns. The revised paper is in significantly better shape. The figures are much clearer and more original, and the (softened) central claim is defensible. Additionally, moving some of the less directly relevant content to the Appendix makes this survey paper more digestible. The distinction between Level 2 and Level 3 in the cognitive hierarchy is reasonable. I still have a soft preference to remove the terms “vehicle-to-environment,” “vehicle-to-agents,” and “vehicle-in-society” as they are quite confusing. I believe the presentation of the hierarchy would be better without these terms. For example, other "agents" are typically considered part of the AD's "environment". Otherwise, I'm happy with the paper in its current form.

---

> ### Author Response · Authors · 2026-02-26
>
> Thank you very much for your thoughtful follow-up and for your constructive feedback throughout this discussion. We sincerely appreciate your careful reading of the revised manuscript, and we are very glad to hear that the paper is now in much better shape and that the main concerns have been addressed.
>
> Thank you as well for this additional suggestion regarding the terms "vehicle-to-environment," "vehicle-to-agents," and "vehicle-in-society." These labels were originally introduced as intuitive cues, but we agree that they may also create unnecessary confusion. We appreciate this point, and we will simplify the presentation of the hierarchy by removing or de-emphasizing these terms in the revised version.
>
> Thank you again for your valuable feedback, which has been very helpful in improving the paper.

---

### Review · Reviewer_3CCa · 2026-01-05

**Summary Of Contributions:**

This long survey paper examines the state of Autonomous Driving (AD) research, particularly the AI reasoning facet of AD. It is not just a survey, but rather a position paper supported by a thorough survey analysis of the literature. Rather than viewing ML models in an AD system as simple classifiers/regressors, the interest here is in the use and development of Multimodal Large Language Models (MLLMs) that can perform explicit reasoning to support vehicle control or long-term planning. The authors propose a “cognitive hierarchy” that spans different levels of intervention for the AI models, from sensorimotor control to planning around other vehicles or humans to broader accounting for the social elements of safe driving (traffic regulations, implicit human intentions, etc.), Under this hierarchy, seven grand research challengers are identified across the three levels. The role and limitations of reasoning are then discussed for each of the challenges, leading to several proposals for promising directions of AD research, Last, the role of benchmarks is discussed with an emphasis on existing data/tools and their shortcomings.

As an outsider to AD, I found this to be a very nice contribution. Insiders would probably appreciate it even more.

Strengths:
- Structure: there is no simple listing of papers in this survey. They are all categorized within the proposed hierarchy with supporting illustrations and tables that situate them relative to one another.
- A clear position: I appreciate the opinionated perspective on offer here. While I cannot vouch for how comprehensive the survey is, I can say that writing it required quite a bit of critical thinking and exploration.
- Limitations and future work suggestions abound in this survey. It will be very helpful to junior researchers looking to advance the state of the art, or non-AD researchers looking to test AI reasoning methods on AD benchmarks.

Weaknesses:
- Length: the survey is a bit too long and at times repetitive. Some disciplined editing could shave off a few pages without affecting the content too much.
- The writing could be improved in places; see Requested Changes for details.
- There is no reflection on the broader societal impact of AD; see Broader Impact Concerns for details.

**Audience:**

Yes

**Audience Explanation:**

Looking at the list of papers accepted to TMLR, I can see multiple that are relevant to AD. However, none seem to be cited in this survey. It might make sense to do so in order to “close the loop” between prior AD research in TMLR, and this survey (if accepted for publication). That being said, there are definitely TMLR readers who would be interested in this paper.

**Broader Impact Concerns:**

There are valid concerns regarding autonomous driving. For instance, it is difficult to allocate responsibility in the event of an accident. Better self-driving cars encourage the manufacturing of more cars, which, even if electric, has an environmental cost to it. Although the authors focus on a technical question, reflecting on the broader implications of developing “optimal” self-driving vehicles would be valuable. I recommend this short read for a discussion of some of these concerns: https://carleton.ca/news/story/self-driving-cars-transportation-woes/

**Claims And Evidence:**

Yes

**Claims Explanation:**

This survey is supported by some 120+ references which I believe counts as accurate evidence.

**Requested Changes:**

Issues with the writing:
- p1, abstract: In the second sentence, you write about "current systems" and "social interactions". What kind of systems are you talking about? AD systems? If so, I am not sure what "social" means in that context.
- p1, intro: "The primary bottleneck is shifting...": the bottleneck in what? In the advancement of AD research? In building better AD systems?
- conclusion: LLMs/MLLMs are qualified as “symbolic”. Are they really? The typical view in AI is that symbolic methods are things like SAT, Constraint satisfaction, AI planning, heuristic search. I wonder whether you can use another high-level descriptor here, or explain your perspective.

Small typos and the like:
- p1: "i.e., a deficit..." --> "namely a deficit..."
- p1: "for the large-scale real-world deployments" --> "for large-scale real-world deployment"
- p2: fig. 1: "intersection wants to go straight" --> "intersection that wants to go straight"; "the vehicle need to slow down" --> "needs to slow down"
- p3: fig. 2: "on the reasoning in..." --> "on reasoning in..."
- p4: "increasingly sophisticated" --> "increasin
- p4: CoT abbreviation unnecessarily defined twice
- p5: “numerous models emerge” —> “numerous models have emerged”; “capabilities expand” —> “capabilities have expanded”
- p6, fig. 3: “Objects Detection” —> “Object Detection”
- p7: “This level represents the ultimate tasks in achieving full autonomy.” —> “This level encompasses tasks that are necessary to achieving full autonomy.”
- p8: “socially aware participant” —> “socially-aware participant”
- p9, fig. 4: The font is way too small; please increase it so that it is legible without much zooming in/squinting.
- p9: “encompass the spectrum of higher-level cognition” —> “encompass a wide spectrum of functions that require higher-level cognition”
- p9: “which necessitates robust” —> “which necessitate robust”
- p10: “a lead vehicles sudden braking” —> “a lead vehicle’s sudden braking”
- p11: “the context-dependency of application” —> “context-dependence”
- p12: “the system need for a robust capability….” —> “the need for system-level, robust, dynamic prioritization among conflicting regulations.”
- p13: “(e.g. cameras,…” —> “(e.g., cameras,…”
- p14, fig. 5: unclear what the numbers/ranges such as “1-3, 7-9..” mean. Additionally, there should be an appendix table providing direct reference to each of the methods in this figure, similar to Table 1. Currently it is difficult to map them to the References.
- p17: “which provides” —> “which provide”
- p17: “pioneer the “glass-box”…” —> “have pioneered the “glass-box”…”

---

> ### Author Response · Authors · 2026-02-14
> **Response to Reviewer 3CCa**
>
> Thank you very much for your encouraging and detailed review. We sincerely appreciate your recognition of the paper's structure, analytical perspective, and its potential value to both AD and broader AI communities. We are also grateful for your thorough and constructive suggestions, which have helped us improve the clarity and completeness of the manuscript.
>
> **Q1:**
> The manuscript is somewhat lengthy and occasionally repetitive; clearer editing could improve conciseness.
>
> **A1:**
> Thank you for this suggestion. We conducted a systematic rereading and revision of the entire manuscript, compressing and rewriting repetitive or redundant content to improve overall conciseness and readability. Specifically, we moved part of Section 2 to the Appendix; removed redundant "Open Questions" in Sections 5.3 and 6.3; and relocated the newly added auxiliary tables in Section 5 as well as the original auxiliary tables in Section 6 to the Appendix. In addition, following your checklist, we corrected all identified spelling, grammatical, and stylistic issues, and performed an additional round of language polishing to further improve clarity.
>
> **Q2:**
> Several expressions lack clarity or precision (e.g., "current systems","social interactions", "primary bottleneck“,and describing LLMs/MLLMs as "symbolic").
>
> **A2:**
> Thank you for your careful comments regarding specific wording issues. We have revised and clarified each point as follows:
>
> **Regarding "current systems" and "social interactions"**
>
> In the abstract, "current systems" refers to existing AD systems. We have explicitly clarified this in the revised manuscript.
>
> The term "social" refers to the social-level cognitive capabilities required for autonomous vehicles as traffic participants, including understanding traffic regulations, implicit human intentions, and social norms. This aspect is discussed in Challenge 7 of Section 4, where we state:
>
> "In mixed-traffic environments, an autonomous vehicle must operate not merely as a law-abiding agent but as a socially intelligent participant."
>
> **Regarding "The primary bottleneck is shifting ..."**
>
> Here, "primary bottleneck" refers to the key limiting factor in progressing toward higher levels of autonomous driving (L4/L5). We have clarified this explicitly in the introduction to remove ambiguity.
>
> **Regarding describing LLMs/MLLMs as "symbolic"**
>
> We agree that, in the classical AI literature, the term "symbolic" typically refers to logic-, planning-, and constraint-satisfaction–based methods. To avoid conceptual ambiguity, we have revised the manuscript to clarify our terminology: we no longer characterize LLMs/MLLMs as "symbolic," and instead use more precise, boundary-aware descriptions (e.g., language-mediated, learned, explicit semantic deliberation). We also adjusted the wording around prior rule-/logic-based approaches accordingly to ensure consistency with standard definitions.
>
> **Q3:**
> The survey does not reference relevant AD-related papers previously published in TMLR.
>
> **A3:**
> Thank you for this suggestion. We agree that this omission needed to be addressed. We have reviewed and incorporated relevant AD-related papers previously published in TMLR, and established clearer connections to existing TMLR research in the relevant discussion sections, thereby better situating our survey within the TMLR research context.
>
> **Q4:**
> The paper lacks reflection on the broader societal impact of autonomous driving.
>
> **A4:**
> Thank you very much for raising this important point. We have added a new section titled "Broader Impact and Societal Considerations" before the references, where we briefly discuss the broader societal implications of autonomous driving systems, including responsibility attribution, environmental impact, and societal challenges associated with large-scale deployment.
>
> This addition aims to complement the technical analysis with real-world considerations and to acknowledge the broader implications of deploying autonomous driving technologies.
>
> Once again, we sincerely thank you for your comprehensive and detailed review. Your suggestions have significantly improved the manuscript in terms of clarity, structural rigor, and overall completeness. All related modifications have been highlighted in blue in the revised version.

---

> > ### Comment · Reviewer_3CCa · 2026-02-17
> > **Good rebuttal**
> >
> > Thank you for the thorough response; glad my review was helpful! I see some missing spaces between words in the final section of the main text, e.g., “scenarioscan”. Otherwise, I am quite satisfied with the revised paper.

---

> > > ### Author Response · Authors · 2026-02-22
> > > **Response to Reviewer 3CCa**
> > >
> > > Thank you for the follow-up and for pointing this out. We apologize for the typographical/formatting issue (e.g., missing spaces such as "scenarioscan") in the final section of the main text. We have corrected these spacing errors and additionally performed a full-pass proofreading/formatting check across the manuscript to eliminate similar issues. We appreciate your careful reading and are glad to hear you are otherwise satisfied with the revision.

---

### Review · Reviewer_1tK1 · 2026-02-01

**Summary Of Contributions:**

This survey examines LLM/MLLM-based approaches to autonomous driving across the full pipeline of perception, planning, and action, with an emphasis on driving reasoning in complex and long-tail scenarios. It introduces a structured taxonomy, framed as a cognitive hierarchy, and summarizes seven key reasoning challenges that limit current systems, mapping representative methods to each challenge. The paper also reviews evaluation practices and benchmarks for assessing reasoning and question-answering capabilities in autonomous driving research.

**Audience:**

Yes

**Audience Explanation:**

Reasoning for autonomous driving has become a central topic to push toward safer, higher-level driving systems. This survey offers a clear entry point for the community by summarizing key challenges and organizing current approaches, helping readers both track existing progress and identify open questions highlighted throughout the paper.

**Broader Impact Concerns:**

I did not identify any issues that need a specific impact statement.

**Claims And Evidence:**

Yes

**Claims Explanation:**

From a survey paper standpoint, this work is clearly organized and easy to follow. Its main contributions are presented in a structured way:

* A cognitive framing of the autonomous driving problem (Section 3). *(I personally found this section a bit hard to digest; I discuss that in the next part.)*
* A comprehensive taxonomy of challenges (Section 4).
* A structured review of prior work (Section 5), followed by an open discussion of remaining gaps (Section 5.3).
* A benchmark overview using the same category labels (Section 6), along with a discussion of what future benchmarks should capture (Section 6.3).

Overall, these components make the survey notably comprehensive and support the authors’ central claims.

**Requested Changes:**

Overall, I do not think the paper requires major revisions. However, one critical point stood out to me:

While the writing is generally strong, I only fully engaged with the paper from Section 4 onward. I struggled to see how the proposed **cognitive hierarchy** in Section 3 connects to the rest of the survey. The hierarchy is presented as a key contribution, and the authors clearly put effort into defining it, but its practical role is not sufficiently grounded in the later sections. In particular, it is unclear how this framework helps structure the challenges, guide system design, or inform evaluation. As a result, readers may not fully understand why this hierarchy matters or how it advances reasoning in autonomous driving.

I would recommend that the authors more explicitly tie the cognitive hierarchy to the subsequent taxonomy, survey of methods, and benchmarks: explaining what problems it helps clarify and how it should influence future research directions.

---

> ### Author Response · Authors · 2026-02-14
> **Response to Reviewer 1tK1**
>
> Thank you very much for your positive and thoughtful evaluation of our work. We sincerely appreciate your recognition of the paper’s organization, taxonomy design, and comprehensive review. We are also grateful for your constructive suggestion regarding the role of the cognitive hierarchy in Section 3.
>
> **Q1:**
> The connection between the cognitive hierarchy in Section 3 and the later taxonomy, method review, and benchmark analysis is not sufficiently explicit, making its practical significance unclear.
>
> **A1:**
> Thank you for raising this important issue. We carefully reflected on the concern that the linkage between the cognitive hierarchy (Section 3) and the subsequent sections was not expressed clearly enough.
>
> In our overall design, the cognitive hierarchy proposed in Section 3 is not intended as an isolated conceptual framework, but rather as the structural foundation for the challenge taxonomy introduced in Section 4. The seven challenges are not listed independently; instead, they are abstracted and distilled from different levels of the cognitive hierarchy, each corresponding to deficiencies in specific layers of cognitive capability.
>
> We initially described the structural role of the cognitive hierarchy in the manuscript. For example, in Section 3, we stated:
> "To map the advanced reasoning capabilities of a central Cognitive Core to concrete operational challenges, a granular deconstruction of the driving task itself is required."
>
> And in the opening sentence of Section 4, we wrote:
>
> "Achieving higher levels of vehicle autonomy requires mastering the complex reasoning demands of both the Egocentric Reasoning (C1–C4) and Social-Cognitive (C5–C7) levels."
>
> To make this structural relationship more explicit, we have revised Figure 4 so that it more clearly illustrates how each category of challenges is derived from different levels of the cognitive hierarchy, thereby reinforcing the overall organizational logic of the paper.
>
> Overall, our revisions focus on clarifying the exposition and making the structural relationships more visually explicit. We have not introduced new theoretical modules or additional conceptual components. Through these improvements, we aim to make the organizational role of the cognitive hierarchy in the manuscript more readily understandable to readers.
>
> Once again, we sincerely thank you for your valuable suggestion, which has significantly helped us improve the clarity of the paper’s structural presentation.

---

> > ### Comment · Reviewer_1tK1 · 2026-02-17
> >
> > Thank you for the detailed response. I realize my previous comment may not have been very clear, so let me try to clarify what I’m looking for.
> >
> > What I’m mainly interested in is how this "cognitive hierarchy" perspective could concretely benefit the community going forward. In particular, I would like to better understand what aspects of model training or system design could be informed by cognitive theory, and how that might translate into practical improvements for autonomous systems.
> >
> > For example, paper [1] incorporates ideas from cognitive science and applies them to introduce a new preference training objective. Since this work is a survey paper, I was hoping to see more discussion on how similar cognitive theories could be combined with existing approaches in this area. It would be helpful to highlight papers where related ideas have already been used in a way that aligns with cognitive theory, even if they were not originally framed in that terminology.
> >
> > If prior work has largely overlooked this angle, then something like "Discussion and Insight" section would be a great place to provide concrete examples of how cognitive hierarchy could realistically be used to improve autonomous driving systems. Some specific scenarios or design directions would make the impact of this perspective much clearer.
> >
> > [1] Ethayarajh, Kawin, et al. "KTO: Model alignment as prospect theoretic optimization." ICML 2024.

---

> > > ### Author Response · Authors · 2026-02-22
> > > **Response to Reviewer 1tK1**
> > >
> > > Thank you for the clarification—this is very helpful. We agree that our previous response mainly emphasized how the cognitive hierarchy structures the survey (hierarchy → challenges → benchmarks), but did not sufficiently explain how this perspective can concretely inform model training or system design going forward.
> > >
> > > In the revised manuscript, we addressed this by adding dedicated discussions in **Sections 5.3 and 6.3（Cognitive Hierarchy-Informed Design/Evaluation Principles）**. In Section 5.3, we discuss how recent methods implicitly strengthen different cognitive levels and distill several design principles informed by cognitive theory. In Section 6.3, we clarify how existing benchmarks already reflect layered cognitive evaluation and discuss how the hierarchy can guide capability-oriented metric decomposition.
> > >
> > > We hope these additions make the practical implications of the cognitive hierarchy clearer and more actionable for future research. We sincerely thank you for prompting us to strengthen this aspect of the paper.

---

> > > > ### Comment · Reviewer_1tK1 · 2026-02-23
> > > >
> > > > Thank you for the additional discussion. The paper now shows a clearer connection to the concept of cognitive hierarchy. I'm interested in seeing how future research might build on this survey to develop more cognitively grounded approaches for autonomous driving.

---

> > > > > ### Author Response · Authors · 2026-02-23
> > > > >
> > > > > Thank you for the encouraging follow-up and for your insightful feedback throughout this discussion. We sincerely appreciate your guidance, which helped us clarify the role and practical implications of the cognitive hierarchy and improve the overall presentation of the survey.

---

### Author Response · Authors · 2026-02-14
**Review Feedback Summary and Manuscript Update Overview**

We are pleased that the reviewers recognized our systematic effort in organizing the field of reasoning in autonomous driving. Several reviewers acknowledged the clarity of the manuscript’s overall structure, as well as the analytical framework constructed around reasoning-related issues. In particular, the taxonomy of seven core challenges grounded in the cognitive hierarchy was regarded as reasonable and integrative, enabling an abstraction and synthesis of existing problems rather than a simple enumeration.
The reviewers also recognized our systematic organization of related methods and benchmarks. The discussion of prevailing trends and remaining gaps at the methodological level, together with the overall analysis of benchmarks, was considered detailed and informative. In particular, Table 1 (Table 2 in the revised manuscript), which provides a consolidated summary of benchmarks, serves as a centralized resource that helps readers quickly grasp the broader landscape of datasets and evaluation settings for reasoning in autonomous driving.
In response to the reviewers' comments, we will conduct a focused round of revisions centered on improving conceptual coherence, moderating our claims, strengthening structural focus, enhancing figure originality, and ensuring citation rigor:
1. Clarify and moderate the "bottleneck" claim, emphasizing the coupling between reasoning and perception/prediction/planning/control;
2. Strengthen the organizational role of the cognitive hierarchy introduced in Section 3 across Sections 4–6;
3. Rewrite and redesign Figure 1;
4. Replace multiple figures previously adapted from prior work with improved, original survey-oriented illustrations;
5. Add a consolidated overview table summarizing autonomous driving methods;
6. Compress background material, remove redundancy, and move core survey content forward;
7. Conduct a comprehensive audit of citations and methodological categorizations;
8. Add a brief discussion on broader societal impact.
We have addressed all reviewer concerns in the revised manuscript, with major changes highlighted in **blue**. We also **added a new appendix** and moved several parts there to streamline the paper's structure.

---

### Decision · Action_Editor_YdkZ · 2026-02-26

**Recommendation:** Accept with minor revision

**Additional Comments:**

The authors need to upload a camera ready version.

**Audience:**

Yes

**Audience Explanation:**

Yes, this paper provide a thorough overview of reasoning in driving systems, which is surely of interest to the TMLR audience.

**Claims And Evidence:**

Yes

**Claims Explanation:**

This is a review paper, so there are no empirical claims. The authors back up their assessments of the field with extensive citations to relevant literature.

---

> ### Author Response · Authors · 2026-03-07
> **Camera-ready Version Submitted**
>
> Thank you for your comments and for handling our submission. We have now uploaded the camera-ready version of the manuscript, incorporating final language revisions and minor improvements.